# From Internal Diagnosis to External Auditing: A VLM-Driven Paradigm for Data-Free Online Backdoor Defense

**Binyan Xu** [1]  **Fan Yang** [1]  **Xilin Dai** [2]  **Di Tang** [3]  **Kehuan Zhang** [1]

## Abstract

Deep Neural Networks (DNNs) remain fundamentally vulnerable to backdoor attacks. Traditional data-free defenses largely operate under the paradigm of internal diagnosis methods like *model repairing* or *input robustness*, yet these approaches are often fragile under advanced attacks as they remain entangled with the victim model's corrupted parameters. We propose a paradigm shift to data-free **External Semantic Auditing**, using universal Vision-Language Models (VLMs) as independent auditors to decouple defense from the compromised model. We introduce **PRISM** (**P**rototype **R**efinement & **I**nspection via **S**tatistical **M**onitoring), which transforms generic VLMs into domain-adaptive gatekeepers purely via online test-time adaptation. PRISM bridges the domain gap through a *Hybrid VLM Teacher* that refines prototypes from the test stream and an *Adaptive Router* that calibrates thresholds via statistical monitoring. Evaluation across 17 datasets and 11 attack types confirms PRISM achieves state-of-the-art performance (suppressing Attack Success Rate to $< 1\%$ on CIFAR-10), proving that robust defense is achievable without touching the model weights or accessing a single training sample. Code: https://github.com/binyxu/PRISM

## 1. Introduction

Deep Neural Networks (DNNs) have achieved remarkable success in critical domains, yet they remain fundamentally vulnerable to *backdoor attacks* (Gu et al., 2019; Chen et al., 2017). By poisoning a small fraction of training data, adversaries implant hidden triggers that hijack model behavior on specific inputs while preserving performance on benign data. The threat has rapidly evolved into an arms race: attacks have transitioned from conspicuous patterns to stealthy dynamic (Nguyen & Tran, 2021), clean-label (Turner et al., 2019), and clean-image (Jha et al., 2024) strategies, making traditional detection increasingly difficult.

In Model-as-a-Service scenarios where defenders lack access to clean training data, data-free test-time defense is the last line of defense. However, current research is constrained by the internal diagnosis paradigm, which exhibits fundamental vulnerabilities. (1) **Model Repairing Methods:** These methods attempt to identify backdoors by inspecting the victim model's internal properties, such as neuron activation traces or Channel Lipschitzness (Zheng et al., 2022). While effective against static attacks, they rely on the assumption that backdoors leave conspicuous statistical footprints. Advanced dynamic attacks explicitly optimize triggers to suppress these internal anomalies, effectively bypassing such defenses. (2) **Input Robustness Methods:** These approaches, including input purification (Shi et al., 2023) and consistency checks (Liu et al., 2023), assume that triggers are sensitive to input perturbations or transformations. However, this assumption fails against *natural* backdoors (e.g., clean-image attacks using physical objects or filters), where the trigger is a robust, semantic feature inherent to the image, indistinguishable from benign features via perturbation. Consequently, a defense that remains robust across diverse attack modalities without relying on the compromised model's internal states or fragile input assumptions remains an open challenge.

To overcome these limitations, we argue for a **paradigm shift**: moving from internal diagnosis to **Online External Semantic Auditing**. Instead of asking the compromised model to verify itself or testing the input's robustness, we introduce an independent, "clean" agent to audit the prediction process. Universal Vision-Language Models (VLMs), such as CLIP (Radford et al., 2021) and Qwen-VL (Bai et al., 2025), offer the ideal foundation for this paradigm due to their vast, open-world semantic knowledge. By decoupling the defense mechanism from the victim model's weights, we theoretically eliminate the attack surface exploited by weight-manipulation attacks.

[1]The Chinese University of Hong Kong, Hong Kong [2]Zhejiang University, China [3]Sun Yat-sen University, China. Correspondence to: Di Tang <tangd9@mail.sysu.edu.cn>, Kehuan Zhang <khzhang@ie.cuhk.edu.hk>.

*Proceedings of the 43rd International Conference on Machine Learning*, Seoul, South Korea. PMLR 306, 2026. Copyright 2026 by the author(s).

While this paradigm holds immense promise, realizing its full potential requires addressing two key challenges: the **domain gap**, where VLMs underperform on specialized tasks compared to the victim model; and **threshold fragility**, where the semantic margin between correct and poisoned predictions varies drastically across samples, making static thresholding unreliable.

We propose PRISM (**P**rototype **R**efinement & **I**nspection via **S**tatistical **M**onitoring), the first comprehensive framework for online external semantic auditing. PRISM wraps the victim model in a dual-stream architecture, employing the VLM not merely as a classifier, but as an evolving *semantic gatekeeper*. (**1**) To bridge the domain gap, we design a **Hybrid VLM Teacher** that augments static text anchors with visual prototypes learned *online* from the test stream. This allows the auditor to adapt to the specific feature distribution of the task purely from the inference stream, thereby circumventing the need for offline training on labeled data. (**2**) To address threshold fragility, we introduce an **Adaptive Router** powered by *online statistical monitoring*. By modeling the logit margin distribution using the Cornish-Fisher expansion, PRISM continuously calibrates the auditing sensitivity, ensuring robustness against distribution shifts and high-noise scenarios (Li et al., 2025a).

Our evaluation across 17 datasets and 11 state-of-the-art backdoor attacks confirms the superiority of this new paradigm. PRISM effectively neutralizes attacks that bypass model repair and input purification methods, such as clean-image and adaptive flooding attacks. It consistently suppresses the Attack Success Rate (ASR) to $< 1\%$ on CIFAR-10 while even improving Clean Accuracy (CA). PRISM establishes that decoupling defense through external semantic auditing is not only feasible but essential for next-generation model security.

Our main contributions are summarized as follows:

- **New Paradigm (External Semantic Auditing):** We identify failure modes of internal diagnosis paradigms and propose a novel data-free direction that uses universal VLMs as independent, external auditors, effectively decoupling defense from compromised victim models.
- **Online Adaptive Framework (PRISM):** We propose PRISM, a test-time framework that turns generic VLMs into effective auditors. With a *Hybrid VLM Teacher* and an *Adaptive Router*, we bridge the domain gap and avoid static thresholds via online adaptation and statistical monitoring.
- **Superior Generalization and Robustness:** PRISM achieves state-of-the-art performance across 17 datasets and 6 VLM backbones. It demonstrates unique robustness against advanced clean-image and adaptive attacks where traditional paradigms fail, setting a new standard for model-agnostic defense.

## 2. Related Work

### 2.1. Vision-Language Models (VLM)

**Vision-Language Embedding Models.** These models bridge modalities by embedding large-scale data into a shared feature space, simultaneously learning visual and textual representations. CLIP (Radford et al., 2021), a seminal VLE model, achieves strong generalization by training on a 400M image-text dataset, treating paired images and texts as positives and all others as negatives.

**Vision-Language Generative Models.** Large Vision-Language Models (LVLMs) extend Large Language Models (LLMs) with visual understanding capabilities. Open-source models like LLaVA (Liu et al., 2024a) connect vision encoders with LLMs via projection layers for visual instruction tuning. More recent advancements include Qwen-2.5-VL (Bai et al., 2025), which excels in high-resolution processing, and Gemma-3 (Team et al., 2025), which adopts a native multimodal architecture. Despite their growing popularity, their potential in enhancing model security, particularly for defending against backdoors, has not been investigated.

**VLM for Security.** Most existing works on VLM focus on security concerns of VLM itself, such as backdoor attacks (Liang et al., 2024), adversarial attacks (Xu et al., 2025a), and pre-training defenses (Yang et al., 2024). The application of VLMs in defending other models remains underexplored. Limited exceptions use VLM for training-time backdoor defense (Xu et al., 2025b; Sabolić et al., 2025) or for mitigating backdoors in Federated Learning (Gai et al., 2025). In distinct contrast, our method addresses a more practical and challenging scenario, representing the first use of VLM for data-free backdoor defense without requiring training interventions or access to model weights.

### 2.2. Backdoor Attacks and Defenses

#### 2.2.1. BACKDOOR ATTACKS.

Backdoor attacks are often implemented via data poisoning (Gu et al., 2019; Chen et al., 2017; Barni et al., 2019), where adversaries inject poison samples into the training dataset, causing the victim model to associate backdoor triggers with target classes. Beyond data poisoning, backdoor attacks can also manipulate the training process (Nguyen & Tran, 2021; 2020; Wang et al., 2022), though these can be adapted into data poisoning by introducing additional noise samples (Wu et al., 2024). Some attacks use benign images as backdoor triggers (Jha et al., 2024; Xu et al., 2026c) to evade detection.

#### 2.2.2. DATA-FREE BACKDOOR DEFENSES.

**Model Repairing Methods** aim to remove backdoors from a compromised model by modifying model weights. Some

approaches use model-intrinsic properties, such as Channel Lipschitzness Pruning (CLP) (Zheng et al., 2022) and its improved version LPP (Wang et al., 2025), which prune neurons based on sensitivity differences. Others utilize model compression techniques to destroy fragile backdoor correlations (Phan et al., 2024; Yang et al., 2026b). Additionally, methods like BCU (Pang et al., 2023) assume access to a set of unlabeled data to purify the model via self-supervised learning. While our method also operates without labeled training data, it distinctively employs an online learning paradigm during testing, continuously adapting to the input stream rather than performing a one-time static repair.

**Input Robustness Methods** focus on identifying or mitigating backdoored inputs with robustness measures during the inference stage. SCALE-UP (Guo et al., 2023) and TeCo (Liu et al., 2023) detect triggers by analyzing prediction consistency under input scaling or corruption. BDMAE (Sun et al., 2023) and ZIP (Shi et al., 2023) employ pretrained generative models (MAE and Stable Diffusion) for zero-shot image purification to remove trigger patterns. More recently, REFINE (Chen et al., 2025) utilizes model reprogramming to learn visual prompts that neutralize backdoors. Although our approach falls into this category, it differs fundamentally from prior works by introducing an *online* mechanism. We are the first to combine the semantic power of Universal VLMs with online statistical monitoring to dynamically refine the defense strategy test-time, offering superior adaptability and robustness.

## 3. Preliminary

### 3.1. Notations.

We consider a classification model $f(\cdot; \theta)$ parameterized by $\theta$. Let $\mathcal{X}$ denote the input space. Let $p(y \mid x; \theta)$ denote the predicted probability assigned by the model $f(\cdot; \theta)$ to class $y$ given input $x$. The predicted label for $x$ is then defined as

$$\hat{y}(x) = \arg\max_{y \in \mathcal{Y}} p(y \mid x; \theta),$$

where $\mathcal{Y} = \{1, 2, \ldots, C\}$ is the set of possible class labels. We define the trigger function as $\mathcal{T} : \mathcal{X} \to \mathcal{X}$, and let $y_t \in \mathcal{Y}$ represent the attack's designated target class. $\mathcal{P}$ denotes the distribution of clean, unaltered samples. The clean accuracy (CA) of the model is the probability of correct classification on clean data, $\mathbb{P}_{(x,y)\sim\mathcal{P}}[\hat{y}(x) = y]$. The attack success rate (ASR), $\mathbb{P}_{(x,y)\sim\mathcal{P}\mid y\neq y_t}[\hat{y}(\mathcal{T}(x)) = y_t]$, measures the probability that a sample $x$ with a trigger is misclassified as the target class $y_t$.

### 3.2. Threat Model.

We adopt a stringent data-free test-time threat model where the defender deploys a potentially compromised "suspicious model" $f_S$ strictly without access to any labeled reference data. The adversary may have embedded arbitrary backdoors—including stealthy clean-label or dynamic triggers—into $f_S$. To secure the inference, the defender utilizes a public, pre-trained, and clean VLM as an external auditor. As a foundation model pre-trained on billion-scale open-world corpora, the VLM's feature space is statistically independent to the specific, poisoned distribution of the victim, making it blind to local triggers. Furthermore, the VLM remains strictly frozen during the defense phase, eliminating the attack surface for test-time weight manipulation. The defense goal is to purify the prediction stream online, minimizing ASR while preserving CA.

## 4. Methodology

### 4.1. Overview of PRISM

To secure deployed models against diverse backdoor attacks without accessing training data, we propose **PRISM** (**P**rototype **R**efinement & **I**nference via **S**tatistical **M**argin). As illustrated in Fig. 1, PRISM functions as a test-time wrapper around the suspicious victim model. The core intuition is that while backdoor triggers force the victim model to output high-confidence predictions on poisoned samples, these predictions often lack semantic support in the feature space of a benign, general-purpose Vision-Language Model (VLM). PRISM exploits this semantic discrepancy through a three-pillar architecture: (1) A **Dual Stream Inference** framework that processes inputs through both the victim model and a trusted Universal VLM Teacher; (2) A **Hybrid VLM Teacher** that enhances zero-shot capabilities by fusing static text anchors with dynamically refined visual prototypes; and (3) An **Adaptive Router** that employs online statistical monitoring to gate predictions based on a difficulty-aware logit margin. By continuously updating the statistical moments and visual prototypes using Cumulative Moving Average (CMA), PRISM adapts to the test stream distribution in an online manner, effectively filtering out backdoor triggers while preserving benign performance.

### 4.2. Dual Stream Inference Framework

The foundation of PRISM is a parallel inference mechanism designed to decouple the potential risk from the decision process. For any incoming test sample $x$, the system initiates two concurrent forward passes. The first stream passes $x$ through the suspicious model $f_S$, yielding a prediction $\hat{y}_S$ and its associated logits. While $f_S$ is potentially compromised, it typically possesses high domain-specific utility. The second stream processes $x$ through a frozen, trusted VLM backbone, as an external auditor like (Zang et al., 2025). By employing a frozen VLM, we mitigate the risk of internal failures in discriminative models, such as attention hacking (Zang, 2025). Unlike prior works that rely solely on static zero-shot classification, which may suffer from do-

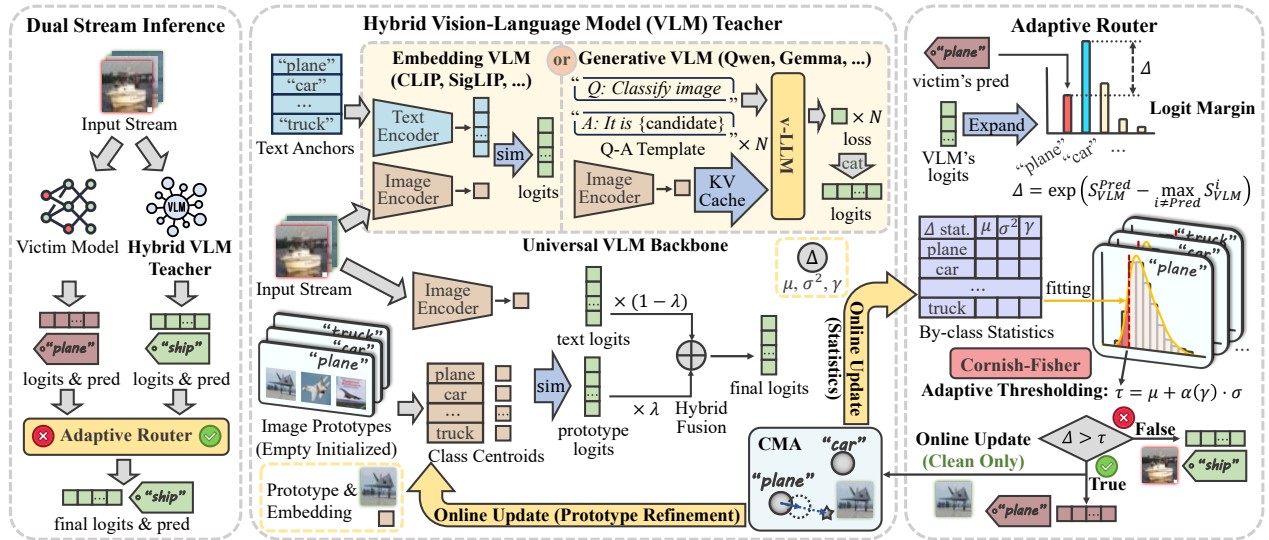

*Figure 1.* **Pipeline of PRISM** (Prototype Refinement & Inspection via Statistical Monitoring).

main shifts, our framework treats the VLM as a "Universal Teacher" that provides a semantic verification signal. The outputs of these two streams—the suspicious logits from the victim and the robust logits from the hybrid VLM—are then fed into the Adaptive Router. This dual-stream design ensures that the system defaults to the high-performance victim model for benign samples but falls back to the robust VLM prediction when a backdoor attack is detected.

### 4.3. Universal Hybrid VLM Teacher

A critical limitation of using standard VLMs for defense is their variable zero-shot performance on specialized downstream tasks. To address this, we introduce a Universal VLM Backbone that supports both embedding-based models (e.g., CLIP, SigLIP) and generative models (e.g., Qwen, Gemma), enhanced by a hybrid anchoring mechanism.

**Universal Backbone Support.** For embedding-based VLMs, we compute the cosine similarity between the image embedding $z_{img}$ and the text embeddings $z_{text}$ of the candidate class labels. For generative VLMs, which are typically computationally intensive, we optimize inference using a Key-Value (KV) Cache strategy. We construct a prompt template $Q$: "Classify this image", and evaluate the probability of the model generating specific class names as answers. By caching the visual features and the system prompt's hidden states, we avoid redundant computations for the invariant parts of the input. Formally, letting $t_k$ be the first sub-word token of class $k$'s label and $P_\phi(\cdot \mid x, Q)$ the VLM's output distribution, the generative logit is $S_{gen}^k = \log P_\phi(t_k \mid x, Q)$, effectively converting generative output into a discriminative probability distribution.

**Hybrid Anchors: Text and Prototypes.** To bridge the gap between general semantic knowledge and task-specific fea-

ture distributions, we employ a hybrid fusion strategy. While static text anchors provide high-level semantic guidance, they often fail to capture fine-grained intra-class variance. Inspired by the success of prototype evolution (Zhu et al., 2025) and CLIP-guided prototype refinement (Tan et al., 2026), we augment the auditor with dynamic visual prototypes (k-class centroids), denoted as $\mathcal{C} = \{c_1, \ldots, c_K\}$. Specifically, given the normalized image embedding $z_{img}$ of an incoming test sample, we compute two distinct logit sets: (1) $S_{text}$, derived from the cosine similarity between $z_{img}$ and the predefined text anchors $\mathcal{T}$; and (2) $S_{proto}$, calculated as the similarity between $z_{img}$ and the current class centroids $\mathcal{C}$. Crucially, to adhere to our data-free constraint, $\mathcal{C}$ is initialized and refined online using samples from the test stream (detailed in Sec. 4.5). The final VLM logits $S_{VLM}$ are obtained via the weighted fusion:

$$S_{VLM} = \lambda \cdot \underbrace{\text{Sim}(z_{img}, \mathcal{C})}_{S_{proto}} + (1 - \lambda) \cdot \underbrace{\text{Sim}(z_{img}, \mathcal{T})}_{S_{text}}, \quad (1)$$

where $\lambda$ is a balancing coefficient. This formulation allows the VLM teacher to use broad linguistic semantics while adapting to the specific visual manifold of the test stream.

### 4.4. Adaptive Router via Statistical Monitoring

Adaptive Router is the core decision logic of PRISM, which determines whether to accept the victim model's prediction or reject it in favor of the VLM. This decision is governed by a statistical discrepancy test based on logit margins, similar to the bias decoupling regularization techniques in natural language inference (Zang & Liu, 2024).

**Logit Margin Calculation.** We define a discrepancy metric $\Delta$ that quantifies the semantic support the VLM provides for the victim model's prediction $\hat{y}_S$. We examine the VLM's logits $S_{VLM}$ specifically at the index $\hat{y}_S$ and compare it to

the maximal logit of all other classes $i \neq \hat{y}_S$. The logit margin $\Delta$ is formally defined as:

$$\Delta = \exp\left(S_{VLM}^{\hat{y}_S} - \max_{i \neq \hat{y}_S} S_{VLM}^i\right) \quad (2)$$

We employ the exponential transformation to ensure statistical stability. Since raw logit differences are unbounded and can approach $-\infty$ when the VLM strongly rejects the victim's prediction, the exponential mapping suppresses these extreme negative outliers, confining the metric to a robust positive range. Consequently, a small $\Delta$ (near 0) implies strong disagreement, while a large $\Delta$ indicates alignment.

**Adaptive Thresholding via Cornish-Fisher Expansion.** A static threshold for $\Delta$ is insufficient due to the non-Gaussian, often skewed nature of the discrepancy distribution (empirical validation in App. F). To address this, we model the distribution of $\Delta$ for each class $k$ online, maintaining the running statistics of mean $\mu_k$, standard deviation $\sigma_k$, and skewness $\gamma_k$. We employ the Cornish-Fisher expansion to derive an adaptive threshold $\tau_k$, which approximates the quantiles of a non-Gaussian distribution (derived in App. A). Let $\zeta$ denote the base confidence coefficient. The skewness-adjusted coefficient $\tilde{\zeta}_k$ is calculated as:

$$\tilde{\zeta}_k = \zeta + \frac{\gamma_k}{6}(\zeta^2 - 1). \quad (3)$$

This correction term shifts the threshold to account for asymmetry: positive skewness raises the threshold to reduce false rejections, while negative skewness lowers it. The final adaptive threshold is defined as $\tau_k = \mu_k + \tilde{\zeta}_k \cdot \sigma_k$. If the discrepancy $\Delta$ of a test sample exceeds $\tau_k$, it is deemed consistent and accepted.

### 4.5. Online Update and Prototype Refinement

To address statistical instability during the initial "cold start" phase, PRISM employs *progressive confidence weighting*. We initialize the router with a *conservative Gaussian prior* (assuming skewness $\gamma = 0$) and linearly interpolate to the skewness-aware Cornish-Fisher correction as the sample count accumulates ($N \approx 100$). This ensures statistical stability before activating the full adaptive mechanism.

After warm-up, we apply a feedback loop that updates internal states only when a sample is certified as clean (i.e., $\Delta > \tau$). Crucially, to ensure robustness against temporal distribution shifts, we employ Cumulative Moving Average (CMA) rather than Exponential Moving Average (EMA). Unlike EMA, which is susceptible to local drift, CMA provides statistical inertia, ensuring that the global distribution estimate becomes increasingly stable over time. Furthermore, this update process is designed for memory efficiency; we strictly update statistical moments ($\mu, \sigma, \gamma$) and class feature centroids ($c_k$) in place without retaining a replay buffer of raw images, thereby minimizing computational overhead to $O(1)$ (algorithm details in App. B).

## 5. Evaluation

### 5.1. Experiment Setup

**Attack Baselines.** We evaluate PRISM against 11 SOTA backdoor attacks in four categories: *Classic* (BadNets (Gu et al., 2019), Blend (Chen et al., 2017)); *Dynamic* (SSBA (Li et al., 2021), IAB (Nguyen & Tran, 2020), WaNet (Nguyen & Tran, 2021), BPP (Wang et al., 2022)); *Clean-Label* (LC (Turner et al., 2019), SIG (Barni et al., 2019), CTRL (Li et al., 2023)); *Clean-Image* (FLIP (Jha et al., 2024), GCB (Xu et al., 2026c)). We set target label $y_t = 0$, using a 50% poison rate for clean-label attacks and 5% otherwise (full config in App. D).

**Defense Baselines.** We compare PRISM with 8 recent SOTA defenses under the identical data-free test-time setting: (1) *Model repairing*: CLP (Zheng et al., 2022), C&C (Phan et al., 2024), LPP (Wang et al., 2025); (2) *Input robustness*: ScaleUp (Guo et al., 2023), ZIP (Shi et al., 2023), TeCo (Liu et al., 2023), BDMAE (Sun et al., 2023), Refine (Chen et al., 2025), with configuration from BackdoorBench (Wu et al., 2022) or their official repositories. We also have two VLM baselines: *Zero-shot*, which directly uses VLMs for classification; and *Ensemble*, which averages the logits of the victim model and VLM for prediction.

**PRISM Configuration.** We evaluate PRISM across diverse VLMs, including embedding models (CLIP (Radford et al., 2021), SigLIP (Zhai et al., 2023), ImageBind (Girdhar et al., 2023)) and generative models (Qwen2.5-VL-7b (Bai et al., 2025), LLaVA-1.5-7b (Liu et al., 2024a), Gemma3-4b (Team et al., 2025)), all using official pre-trained weights. Unless otherwise specified, we set the base adaptive threshold $\zeta = -2$, victim model to PreActResNet18, Auditing VLM to CLIP, batch size to 256, and prototype fusion weight $\lambda_p = 0.5$. For warm-up, we use the initialization window $N$ of only one batch of unlabeled images.

**Datasets.** Our evaluation spans 17 datasets categorized into: (1) *General Datasets* (e.g., CIFAR-10, CIFAR-100, ImageNet), and (2) *Domain-Specific Datasets* (e.g., MNIST, GTSRB, Texture datasets). The latter poses a significant challenge, as general VLMs often exhibit significant performance gaps compared to specialized victim models (e.g., 40% VLM accuracy vs. 95% victim accuracy on MNIST).

### 5.2. Main Results

#### 5.2.1. PRISM VS. BASELINES.

**Performance Superiority.** PRISM significantly outperforms all existing data-free and test-time baselines. As detailed in Table 1, PRISM is the sole method capable of suppressing ASR below 15% across all 11 attack types while simultaneously improving Clean Accuracy (CA) on CIFAR-10. Existing SOTA methods exhibit severe vulner-

*Table 1.* **Comparative evaluation of backdoor defenses on CIFAR-10.** ASRs below 15% are highlighted in blue (success), while those above 15% are in red (failure). PRISM is the only method that successfully defends against all 11 attack types while maintaining high Clean Accuracy (CA), whereas existing SOTA defenses fail significantly on Clean-Image backdoors (FLIP, GCB) or Dynamic attacks.

| Defense → | No Defense | | CLP | | ScaleUp | | BDMAE | | ZIP | | TeCo | | C&C | | LPP | | Refine | | Zero-shot | | Ensemble | | PRISM (Ours) | |
|---|---|---|---|---|---|---|---|---|---|---|---|---|---|---|---|---|---|---|---|---|---|---|---|---|
| | | | | | | | | | | | | | | | | | | | Naive VLM Baselines | | | | | |
| Attack ↓ | CA | ASR | CA | ASR | CA | ASR | CA | ASR | CA | ASR | CA | ASR | CA | ASR | CA | ASR | CA | ASR | CA | ASR | CA | ASR | CA | ASR |
| **Classic** BadNet | 92.3 | 87.9 | 90.7 | 12.4 | 90.2 | 0.2 | 90.0 | 0.1 | 86.4 | 9.1 | 92.4 | 26.4 | 90.3 | 9.5 | 90.7 | 3.3 | 90.7 | 1.7 | 86.7 | 0.2 | 94.9 | 50.4 | 93.9 | 0.0 |
| Blend | 93.6 | 99.4 | 90.3 | 76.3 | 84.1 | 97.5 | 91.6 | 73.4 | 87.3 | 5.7 | 83.8 | 0.8 | 84.7 | 50.7 | 83.9 | 15.9 | 91.6 | 2.9 | 86.7 | 0.2 | 95.5 | 95.3 | 94.2 | 0.0 |
| **Dynamic** WaNet | 91.1 | 94.1 | 89.1 | 2.0 | 76.2 | 89.2 | 89.5 | 9.7 | 45.6 | 70.9 | 92.0 | 99.0 | 91.1 | 94.0 | 89.0 | 43.3 | 82.9 | 2.2 | 86.7 | 0.3 | 95.0 | 59.8 | 93.6 | 0.0 |
| BPP | 91.5 | 99.4 | 91.2 | 1.7 | 85.3 | 0.4 | 88.5 | 38.0 | 83.8 | 2.2 | 88.7 | 10.9 | 90.4 | 86.1 | 91.3 | 1.9 | 90.3 | 1.1 | 86.7 | 0.4 | 94.6 | 95.2 | 93.7 | 1.1 |
| IAB | 91.5 | 95.0 | 91.1 | 4.5 | 82.9 | 2.3 | 89.8 | 40.5 | 82.3 | 61.3 | 80.8 | 7.9 | 86.0 | 25.4 | 90.5 | 2.9 | 91.1 | 6.0 | 86.7 | 0.5 | 94.4 | 17.9 | 92.4 | 0.0 |
| SSBA | 93.0 | 97.3 | 90.7 | 42.8 | 83.7 | 0.0 | 91.5 | 15.3 | 88.1 | 10.4 | 84.3 | 4.5 | 84.0 | 13.0 | 90.5 | 29.4 | 60.9 | 5.9 | 86.7 | 0.3 | 95.1 | 88.0 | 93.8 | 0.0 |
| **Clean Label** CTRL | 93.6 | 95.9 | 91.7 | 0.9 | 82.4 | 43.1 | 92.1 | 19.7 | 88.3 | 1.6 | 85.3 | 11.8 | 93.1 | 95.3 | 92.7 | 36.1 | 91.1 | 4.8 | 86.7 | 0.4 | 95.7 | 75.7 | 94.2 | 0.0 |
| SIG | 93.6 | 93.9 | 90.3 | 93.5 | 84.5 | 79.3 | 92.3 | 79.9 | 90.0 | 97.3 | 83.5 | 6.2 | 85.7 | 34.6 | 92.4 | 93.2 | 90.5 | 0.3 | 86.7 | 0.7 | 95.4 | 80.7 | 92.6 | 2.2 |
| LC | 93.4 | 98.4 | 87.4 | 18.4 | 88.2 | 0.0 | 92.8 | 0.0 | 89.7 | 1.0 | 84.2 | 7.2 | 85.4 | 30.9 | 89.4 | 12.2 | 91.5 | 2.0 | 86.7 | 0.3 | 95.5 | 73.5 | 94.6 | 0.0 |
| **Clean Image** FLIP | 89.9 | 99.2 | 89.1 | 28.5 | 81.5 | 0.1 | 88.3 | 94.5 | 86.5 | 87.6 | 85.3 | 50.0 | 87.3 | 69.1 | 82.2 | 9.6 | 88.8 | 30.3 | 86.7 | 0.2 | 93.8 | 96.3 | 91.7 | 1.1 |
| GCB | 88.6 | 100. | 89.1 | 100. | 81.5 | 100. | 85.6 | 100. | 78.7 | 100. | 88.9 | 100. | 86.9 | 100. | 86.4 | 100. |  |  | 86.7 | 0.4 | 92.3 | 100. | 90.1 | 4.4 |
| Average | 92.0 | 96.4 | 90.1 | 34.6 | 83.7 | 37.5 | 90.2 | 42.8 | 82.4 | 40.6 | 86.3 | 29.5 | 87.7 | 55.3 | 89.3 | 28.8 | 86.9 | 14.3 | 86.7 | 0.4 | 94.7 | 75.7 | 93.2 | 0.8 |
| CA Drop (smaller is better) | | | ▼1.9 | | ▼8.3 | | ▼1.8 | | ▼9.6 | | ▼5.7 | | ▼4.3 | | ▼2.7 | | ▼5.1 | | ▼5.3 | | ▲2.7 | | ▲1.2 | |
| ASR Drop (larger is better) | | | | ▼61.8 | | ▼58.9 | | ▼53.6 | | ▼55.8 | | ▼66.9 | | ▼41.1 | | ▼67.6 | | ▼82.1 | | ▼96.0 | | ▼20.7 | | ▼95.6 |

abilities: defenses like CLP and ScaleUp fail completely against Clean-Image backdoors (FLIP, GCB), while ZIP and TeCo struggle with Dynamic attacks. Notably, the strongest baseline, Refine (Chen et al., 2025), fails to defend against Clean-Image attacks. While the naive Zero-shot VLM baseline achieves low ASR, it incurs catastrophic CA drops (e.g., $\geq 70\%$ drop on GTSRB), rendering it impractical. PRISM successfully bridges this gap, offering comprehensive protection without compromising utility.

**Overhead Efficiency.** Fig. 2 illustrates the runtime performance on an NVIDIA A100 GPU with a batch size of 1. PRISM (CLIP) shows high efficiency, ranking as high as *2nd* among defenses with an inference latency of *12.5 ms* per image. While large generative backbones (e.g., Qwen) will incur higher latency compared to embedding models, our extensive evaluations confirm that the lightweight CLIP auditor achieves sufficient defense performance, making the additional computational cost of generative models unnecessary for most practical deployment scenarios.

### 5.3. Robustness Evaluation

#### 5.3.1. ROBUSTNESS ACROSS VLM ARCHITECTURES.

PRISM exhibits strong robustness to VLM architectural variations. As shown in Fig. 3, whether using embedding models (CLIP, SigLIP) or generative models (Qwen, LLaVA), PRISM consistently limits ASR to below 10% across both general and domain-specific datasets (expanded results in App. L). A *counter-intuitive observation* is that PRISM maintains high CA even when VLM's zero-shot accuracy is poor (e.g., on GTSRB). Conversely, when zero-shot accuracy is higher (e.g., ImageBind on TinyImageNet), PRISM can even repair CA to match superior zero-shot performance. This phenomenon occurs because our *Adaptive Router* effectively decouples defense capability from VLM performance: in low-performance domains, the VLM acts as

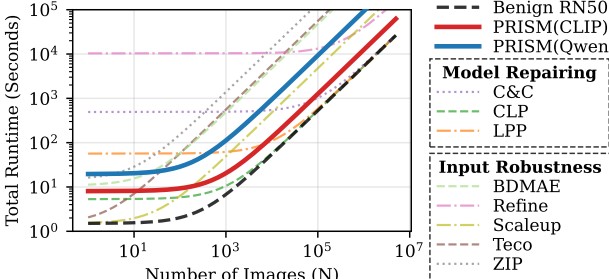

*Figure 2.* **Inference Latency comparison for all data-free defenses on ImageNet.** PRISM (CLIP version) ranks 2/9 at best and ranks 4/9 at worst among all the tested methods.

a weak constraint that only intercepts high-confidence statistical outliers (triggers), while deferring to the victim model's superior domain knowledge for benign samples. For specialized domains like healthcare and backdoored VLMs, we further explore a "Trust Chain" strategy in App. E.

#### 5.3.2. DATASET SIZE SCALABILITY.

PRISM demonstrates robustness across dataset sizes varying by three orders of magnitude. Figure 4 shows that ASR is consistently suppressed below 20% across all scales. We observe a performance pattern related to the ratio of poisoned to clean samples (upper triangle region): when the number of backdoor samples significantly exceeds clean samples, performance slightly degrades. This is expected, as the online statistical monitoring is influenced by the dominant data mode. However, even in extreme scenarios where the poison set is $50\times$ larger than the clean set, ASR remains effectively controlled at $\sim 12\%$, validating the inertia and robustness of our CMA-based online update mechanism.

#### 5.3.3. DATASET SCALABILITY.

Table 2 validates PRISM across 17 datasets with varying scales and Imbalance Ratios (IR, defined as the ratio of samples in the majority class to the minority class). The method

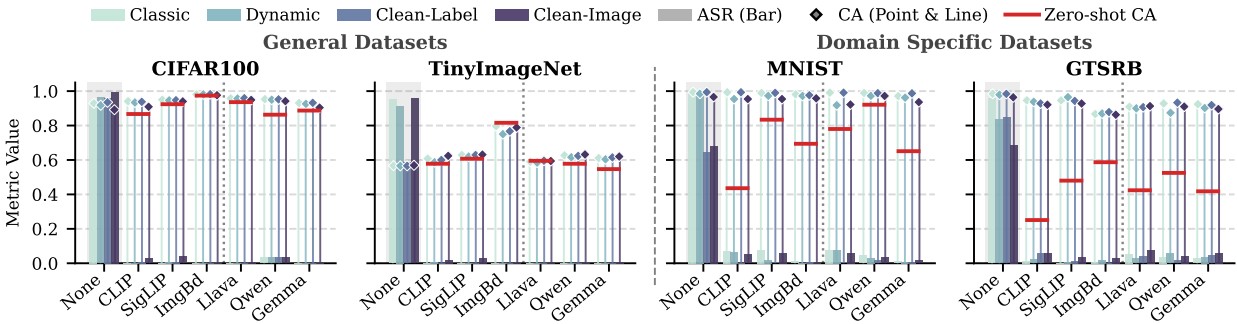

*Figure 3.* **Robustness across VLM Architectures.** PRISM consistently reduces ASR to $< 8\%$ across 6 different VLMs (Embedding & Generative). It maintains CA drops within 5%, even on domain-specific datasets (GTSRB) where baseline VLM performance is low.

*Table 2.* **Scalability across 17 datasets with different Imbalance Ratio (IR).** PRISM successfully defends against all tested attacks (Blend, BPP, CTRL) across all datasets.

| Dataset ↓ | IR | Accuracy | | Blend | | BPP | | CTRL | |
|---|---|---|---|---|---|---|---|---|---|
| | | VLM | RN50 | ΔCA | ASR | ΔCA | ASR | ΔCA | ASR |
| SVHN | 3.20 | 13.4 | **95.5** | ▼1.6 | 0.8 | ▼1.9 | 1.6 | ▼1.8 | 1.8 |
| Country211 | 1.00 | 17.2 | 6.8 | ▲5.3 | 0.0 | ▲5.4 | 1.4 | ▲5.3 | 0.0 |
| GTSRB | 12.5 | 25.1 | **97.8** | ▼4.9 | 2.5 | ▼0.7 | 0.0 | ▼5.9 | 6.7 |
| FER2013 | 16.0 | 41.4 | 63.0 | ▼5.6 | 0.0 | ▼3.8 | 4.8 | ▼4.7 | 4.8 |
| DTD | 1.00 | 43.6 | **99.5** | ▼0.3 | 6.7 | ▲0.0 | 3.3 | ▼0.3 | 0.0 |
| TinyImgNet | 1.00 | 44.3 | 25.7 | ▲17.8 | 0.0 | ▲16.3 | 0.0 | ▲19.6 | 0.0 |
| StanfordCars | 2.83 | 57.8 | 57.2 | ▲1.9 | 0.0 | ▼0.9 | 0.0 | ▲1.8 | 0.0 |
| ImageNet | 1.00 | 59.6 | 49.3 | ▲9.9 | 0.0 | ▲13.8 | 2.5 | ▲5.4 | 0.0 |
| SUN397 | 37.7 | 62.2 | 66.1 | ▼1.2 | 2.0 | ▼1.2 | 0.0 | ▼2.1 | 0.0 |
| MNIST | 1.00 | 62.3 | 69.2 | ▲2.2 | 0.0 | ▲3.4 | 0.0 | ▲1.0 | 0.0 |
| CIFAR100 | 1.00 | 62.5 | 55.3 | ▲12.9 | 0.9 | ▲14.2 | 0.9 | ▲13.0 | 0.9 |
| Flowers102 | 1.00 | 66.5 | 31.0 | ▲15.2 | 0.0 | ▲12.7 | 0.0 | ▲17.7 | 0.0 |
| Caltech101 | 86.5 | 81.6 | 68.8 | ▲12.7 | 0.0 | ▲14.1 | 0.0 | ▲11.3 | 0.0 |
| CIFAR10 | 1.00 | 86.7 | 92.9 | ▲0.5 | 0.0 | ▲2.3 | 1.1 | ▲0.6 | 0.0 |
| OxfordIIITPet | 1.14 | 87.3 | 26.6 | ▲1.9 | 2.9 | ▲5.4 | 2.9 | ▲4.5 | 2.9 |
| Food101 | 1.00 | 88.8 | 68.7 | ▲8.9 | 1.6 | ▲10.9 | 0.0 | ▲14.1 | 0.0 |
| STL10 | 1.00 | **97.1** | 64.8 | ▲33.4 | 1.4 | ▲33.0 | 0.0 | ▲26.2 | 0.0 |

proves robust against extreme imbalance (e.g., Caltech101, IR=86.5) and does not rely on high VLM performance. For instance, despite a low zero-shot accuracy of 13.4% on SVHN, PRISM remains effective (See App. J for TPR/FPR analysis). This implies that even imperfect VLM signals are sufficient to identify trigger-induced statistical anomalies distinct from natural data manifolds.

### 5.3.4. POISON RATE ROBUSTNESS.

Unlike many defenses that are sensitive to poison rates (Zheng et al., 2022), PRISM exhibits strong robustness due to its reliance on semantic logit similarity rather than intrinsic model properties. PRISM controls CIFAR-10 ASR to $\le 5\%$ and GTSRB ASR to $\le 12\%$ across all poison rates (1%, 5%, 10%), demonstrating high practical reliability (additional rates in App. K).

### 5.4. Ablation Study

#### 5.4.1. COMPONENT ANALYSIS

We conduct an ablation study to quantify the contribution of three core components: Online Update, Prototype Refine-

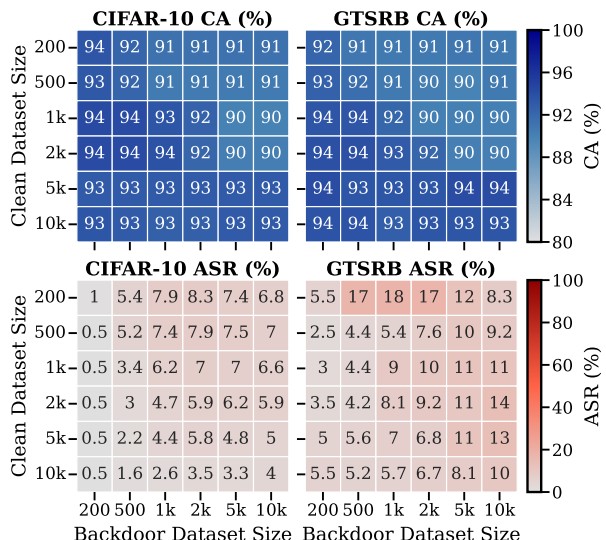

*Figure 4.* **Scalability to Dataset Size.** PRISM maintains stable performance (CA variation $< 4\%$, ASR $< 20\%$) across datasets ranging from 200 to 10,000 images, demonstrating robustness to the scale of the inference stream.

ment, and Skewness Correction. Table 3 summarizes the results (full breakdown in App. M). (1) **Online Update**: This is crucial for adaptation. Without online updates, PRISM fails on domain-specific datasets like GTSRB (ASR spikes to $\sim 40\%$). This failure stems from the inherent bias in pretrained VLMs; static gating propagates this bias, whereas online adaptation corrects it. (2) **Prototype Refinement**: This component significantly impacts Clean Accuracy, particularly on specialized domains. For instance, removing prototypes causes a CA drop of nearly 20% on GTSRB, rendering the defense too costly in terms of utility. (3) **Skewness Correction**: While the aggregated results appear stable, this stability assumes optimal hyperparameter tuning. As shown in Figure 5, PRISM achieves global optimality at $\zeta = -2$ across all datasets. In contrast, without Skewness Correction (i.e., assuming a Gaussian distribution), the optimal threshold varies wildly across datasets (e.g., -2 for CIFAR vs. -3 for GTSRB), making deployment impractical. Skewness Correction effectively standardizes the distribution, allowing for a uniform hyperparameter configuration.

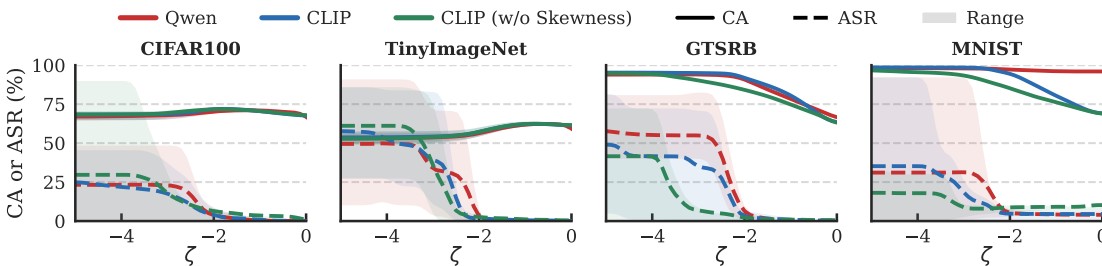

*Figure 5.* **Hyperparameter Sensitivity of** $\zeta$**.** PRISM reaches optimal performance consistently at $\zeta = -2$ across all datasets and attacks. Without Skewness Correction, the optimal threshold fluctuates significantly between datasets, validating the necessity of our Cornish-Fisher expansion for robust, dataset-agnostic deployment.

*Table 3.* **Component Ablation Study.** Impact of removing Online Update, Prototype Refinement, and Skewness Correction. Best results under each case are presented with optimal hyperparameters.

| Ablation Cases→ | Baseline CA | ASR | w/o Online CA | ASR | w/o Prototype CA | ASR | w/o Skewness CA | ASR |
|---|---|---|---|---|---|---|---|---|
| | | | | CIFAR100 | | | | |
| Classic | 72.8 ▲2.3 | 0.0 | 71.5 ▲1.0 | 5.7 | 71.0 ▲0.5 | 1.0 | 72.5 ▲2.0 | 0.0 |
| Dynamic | 70.4 ▲4.5 | 0.0 | 70.4 ▲4.5 | 4.0 | 69.7 ▲3.8 | 2.0 | 70.6 ▲4.7 | 0.8 |
| C-Label | 73.2 ▲2.5 | 0.0 | 72.0 ▲1.4 | 5.1 | 71.1 ▲0.5 | 0.7 | 72.9 ▲2.3 | 0.0 |
| C-Image | 69.1 ▼1.0 | 1.0 | 69.0 ▼1.2 | 6.3 | 68.9 ▼1.3 | 5.6 | 68.3 ▼1.9 | 1.0 |
| | | | | GTSRB | | | | |
| Classic | 94.4 ▼4.1 | 1.3 | 90.3 ▼8.2 | 32.9 | 86.1 ▼12.4 | 3.7 | 94.2 ▼4.3 | 0.8 |
| Dynamic | 92.1 ▼5.8 | 2.1 | 87.9 ▼10. | 21.0 | 86.1 ▼11.8 | 9.6 | 90.2 ▼7.7 | 1.5 |
| C-Label | 92.5 ▼5.9 | 5.8 | 88.2 ▼10. | 31.1 | 83.2 ▼15.3 | 12.0 | 92.7 ▼5.8 | 4.2 |
| C-Image | 90.3 ▼6.1 | 4.5 | 85.3 ▼11. | 8.6 | 80.5 ▼16.0 | 5.8 | 90.0 ▼6.4 | 5.3 |

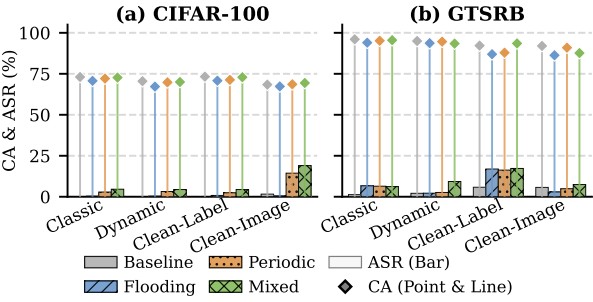

*Figure 6.* **Robustness against Adaptive Attacks.** ASR remains below 20% against 3 types of online adaptive attacks (Flooding, Periodic, Mixed), proving the resilience of the update mechanism.

### 5.4.2. HYPERPARAMETER ANALYSIS

We analyze the base threshold coefficient $\zeta$. Results across two VLM backbones and four datasets confirm that $\zeta = -2$ is consistently optimal. This value is not arbitrary but aligns with statistical theory: in a standard normal distribution, $\mu - 2\sigma$ covers $\approx 95\%$ of the probability mass. Our skewness correction effectively maps the skewed margin distribution to this standard form, allowing the $\zeta = -2$ setting to serve as a robust, theoretically grounded default that generalizes across diverse attack and data landscapes (see Figure 5).

### 5.5. Adaptive Attacks

We evaluate PRISM against adaptive adversaries aware of our defense strategy, considering both online manipulation and semantic evasion.

### 5.5.1. ONLINE ADAPTIVE ATTACKS

We synthesize three adaptive online strategies: (1) *Flooding*, where backdoor samples dominate the stream ($> 90\%$); (2) *Periodic*, employing alternating waves of clean and poisoned data; and (3) *Mixed*, injecting poison from the very start to corrupt the initialization. As shown in Figure 6 and 7, PRISM remains robust against all three strategies (ASR $< 20\%$, CA drop $< 5\%$). This resilience is attributed to our **Selective Update Mechanism**: the system only updates statistics using samples that pass the gate ($\Delta > \tau$). Consequently, flooding attacks are rejected and do not corrupt the internal state. Similarly, for mixed attacks, the statistical inertia provided by the CMA mechanism ensures that minor initial impurities do not destabilize the defense trajectory.

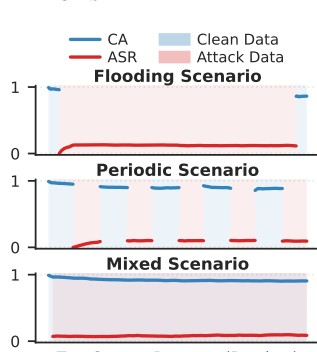

*Figure 7.* **Online Attack Stream Analysis on GTSRB.** The defense remains effective throughout the iteration stream.

### 5.5.2. AUDITOR-AWARE ADAPTIVE ATTACKS

To rigorously stress-test the "trusted auditor" assumption, we evaluate PRISM against semantic typography attacks designed to mislead the VLM. We explicitly exclude visually semantic-consistent triggers from this setting (e.g.: add dog's feature to make images classified as a dog). Such triggers fundamentally alter the visual ground truth of the input, violating the definition of a backdoor attack (where the input should not semantically belong to the target class). Thus, we focus on textual triggers which induce semantic shortcuts without overriding visual features.

As shown in Fig. 9, we superimpose target class names (e.g., "Plane") onto benign images (setup details in App. G). Results in Table 4 reveal that this attack compromises the stan-

*Table 4.* **Defense against adaptive text prompt attack.**

| Prompt Attack | Zero-shot CA | ASR | PRISM CA | ASR |
|---|---|---|---|---|
| Original | 86.7 | 0.2 | 92.3 | 1.3 |
| Adaptive | 86.7 | 60.1 | 92.2 | 15.6 |

*Table 5.* **Auditor Backdoor Robustness under VLM Collusion Scenarios.** Only cross-model collusion breaks our defense.

| Scenario | VLM Compromise | CA (%) | ASR (%) |
|----------|----------------|--------|---------|
| Sc0 | None (clean CLIP) | 93.9 | 0.2 |
| Sc1 | Different trigger | 93.6 | 0.8 |
| Sc2 | Same trigger, different target | 93.9 | 0.0 |
| Sc3 | Same trigger & same target | 93.7 | 100 |

dalone VLM, raising zero-shot ASR to 60.1%. While freezing the VLM eliminates parameter-level tampering, we acknowledge that the *input surface* remains susceptible to semantic manipulation, as evidenced by PRISM's ASR rising to 15.6%. However, PRISM still suppresses the attack significantly compared to the unprotected VLM. This shows that even when the VLM's textual alignment is bypassed, PRISM's *online visual prototype refinement* provides a critical safety layer by detecting the distributional discrepancy between the typographic attack and authentic target samples. Additional feature mixing attacks are discussed in App. H.

### 5.5.3. AUDITOR BACKDOOR ROBUSTNESS

A more fundamental threat is whether the auditing VLM itself can be compromised. We analyze four collusion scenarios (Sc0–Sc3) of escalating severity based on how the attacker coordinates the VLM backdoor with the victim backdoor (CIFAR-10, BadNets; PRISM CA/ASR in %).

Table 5 reveals that PRISM's robustness depends on the coordination structure, not merely the presence of a VLM backdoor. Sc1 (different trigger) and Sc2 (same trigger, different target) cause negligible degradation (≤0.8% ASR), because a VLM misdirected toward a different class actively *assists* the defense against the victim's actual target. Only full collusion (Sc3) breaks the defense, which requires the attacker to simultaneously embed the same trigger toward the same target in both an independently distributed, widely-audited foundation model and the victim model. We argue Sc3 requires a qualitatively stronger adversarial capability than any existing backdoor attack, as it demands coordinated supply-chain compromise of two disjoint model pipelines.

To address the potential compromise in Sc3, we refer to the **Trust Chain strategy** (App. E). This approach requires only a single trusted root node, which can then propagate trust to other domain-specific or foundation models.

## 6. Discussion

PRISM's security guarantee rests on the statistical independence of the auditing VLM from the victim model. This guarantee weakens in two scenarios. First, for domain-specific tasks (e.g., 3D Perception (Ke et al., 2025), industrial state estimation (Dai et al., 2025b; 2026)), publicly available VLMs may lack sufficient zero-shot coverage, reducing audit signal quality; we recommend the

Trust Chain strategy (App. E) as a mitigation. Second, full VLM collusion (SC3 in Sec. 5.5.3), where an attacker poisons both the victim and the auditing VLM with coordinated targets, can degrade performance. However, this attack requires simultaneous supply-chain control over two independently distributed models, which is a qualitatively stronger threat model than standard backdoor attacks. Additionally, PRISM's prototype refinement assumes that test streams contain sufficient clean samples to initialize uncontaminated centroids; the "dirty cold-start" robustness of text anchors provides transient protection during this window (App. P). Finally, extending PRISM to other tasks, such as probabilistic forecasting (Dai et al., 2025a), speech separation (Li et al., 2026a), or code generation (Li et al., 2025d), requires adapting the logit-margin gating to sequence-level confidence; we show a preliminary design in App. S. More broadly, the external auditing paradigm can be further generalized by integrating orchestration for reliable inference (Qiu et al., 2026; Lin et al., 2025), potentially securing LLMs against advanced backdoor threats. For Ethical Considerations see App. T; for Potential Mitigations see App. U.

## 7. Conclusion

We present PRISM, a test-time defense framework redefining backdoor mitigation via online external semantic auditing. Using Vision-Language Models, PRISM overcomes the latency and domain fragility limitations of prior data-free methods. To ensure robust detection without compromising clean accuracy, we design the Hybrid VLM Teacher for prototype refinement and Adaptive Router for statistical gating. Extensive evaluations confirm PRISM neutralizes SOTA attacks, achieving negligible ASRs with superior efficiency. As a model-agnostic solution, PRISM secures deployed models against evolving threats, charting a new direction for multimodal intelligence in trusted AI.

## Impact Statement

PRISM presents a novel test-time defense framework that enhances the security and reliability of DNNs against evolving backdoor threats. By shifting the defense paradigm from internal model diagnosis to external semantic auditing via VLMs, PRISM provides a robust safeguard for Model-as-a-Service deployments where access to training data or model parameters is restricted. The broader impact lies in fostering trust in third-party black-box models: by neutralizing stealthy attack variants including clean-image and dynamic attacks, PRISM mitigates operational risks in safety-critical domains such as healthcare, autonomous driving, and financial systems. Our approach of using foundation models as independent auditors establishes a scalable pathway for AI safety monitoring, encouraging the responsible integration of multimodal intelligence into real-world applications.

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

# A. Theoretical Foundations of Adaptive Thresholding

In the methodology, we introduced an adaptive thresholding mechanism to handle the non-Gaussian distribution of the logit margin $\Delta$. While the Central Limit Theorem suggests that distributions approach normality asymptotically, the finite sample distributions of neural network logits—especially after the exponential transformation in Eq. (2)—often exhibit significant asymmetry (skewness). To address this, we employ the **Cornish-Fisher Expansion**, a moment-based approximation technique, rather than assuming a strict parametric distribution.

## A.1. Edgeworth and Cornish-Fisher Expansions

Let $X$ be a random variable representing the logit margin $\Delta$ for a specific class, characterized by its mean $\mu$, variance $\sigma^2$, and skewness $\gamma$. We define the standardized variable $Z = (X - \mu)/\sigma$. The *Edgeworth expansion* provides a refinement to the standard normal approximation by adding correction terms based on higher-order cumulants. The probability density function (PDF) of the standardized variable $Z$ can be approximated as:

$$f_Z(z) \approx \phi(z) \left[ 1 + \frac{\gamma}{6} H_3(z) \right], \tag{4}$$

where $\phi(z)$ is the standard normal PDF, and $H_3(z) = z^3 - 3z$ is the third Hermite polynomial.

However, for outlier detection and thresholding, we are strictly interested in the *quantiles* of the distribution rather than the density. The *Cornish-Fisher expansion* inverts the Edgeworth expansion to express the quantile $w_\alpha$ of the non-normal distribution $Z$ in terms of the quantile $z_\alpha$ of the standard normal distribution (where $\Phi(z_\alpha) = \alpha$).

The first few terms of the Cornish-Fisher expansion for the $p$-th quantile are given by:

$$w_p \approx z_p + \frac{1}{6}(z_p^2 - 1)\gamma + O(n^{-1}), \tag{5}$$

where:

- $w_p$ is the approximate quantile of the standardized margin distribution.

- $z_p$ is the standard normal quantile (corresponding to our hyperparameter $\zeta$).

- $\gamma$ is the skewness of the margin distribution.

## A.2. Derivation of the Threshold Formula

In PRISM, we seek a threshold $\tau$ that separates benign samples (high consistency) from potential attacks (low consistency) with a confidence level determined by the base Z-score $\zeta$. Mapping back from the standardized domain to the original domain $X \sim \mathcal{D}(\mu, \sigma, \gamma)$, the threshold $\tau$ is derived as:

$$
\begin{aligned}
\tau &= \mu + w_p \cdot \sigma \\
&\approx \mu + \left[ \zeta + \frac{\gamma}{6}(\zeta^2 - 1) \right] \cdot \sigma.
\end{aligned} \tag{6}
$$

Let the skewness-adjusted coefficient be $\tilde{\zeta} = \zeta + \frac{\gamma}{6}(\zeta^2 - 1)$. This directly corresponds to the implementation in our `Adaptive Router`:

$$\tau_{adaptive} = \mu + \tilde{\zeta} \cdot \sigma. \tag{7}$$

**Applicability and Robustness.** This derivation demonstrates that our correction term is not a heuristic but a first-order statistical approximation that adjusts the acceptance region based on the asymmetry of the discrepancy distribution. For a distribution with a heavy left tail (negative skewness), $\tilde{\zeta}$ decreases, effectively lowering the threshold to avoid rejecting valid benign samples. To ensure numerical stability during the online estimation process—where extreme outliers might temporarily distort moments—we strictly clamp the estimated skewness $\gamma$ to the range $[-10, 10]$. This prevents the threshold from diverging due to finite-sample volatility.

# B. Online Estimation of Statistical Moments

PRISM employs a memory-efficient online mechanism to update the statistical moments (mean, variance, skewness) of the margin $\Delta$ without storing historical data. This ensures $O(1)$ time complexity and minimal memory footprint.

Let $S_{1,n}$, $S_{2,n}$, and $S_{3,n}$ denote the sums of the first, second, and third powers of the incoming data stream up to step $n$, respectively. For a new batch of samples $\mathbf{x} = \{x_1, ..., x_m\}$ at step $t$, we update the raw moments as follows:

$$N_t = N_{t-1} + m \tag{8}$$

$$S_{1,t} = S_{1,t-1} + \sum_{i=1}^{m} x_i \tag{9}$$

$$S_{2,t} = S_{2,t-1} + \sum_{i=1}^{m} x_i^2 \tag{10}$$

$$S_{3,t} = S_{3,t-1} + \sum_{i=1}^{m} x_i^3 \tag{11}$$

The expected values (raw moments) are estimated as:

$$E[X] = \frac{S_{1,t}}{N_t}, \quad E[X^2] = \frac{S_{2,t}}{N_t}, \quad E[X^3] = \frac{S_{3,t}}{N_t}. \tag{12}$$

To compute the Cornish-Fisher threshold, we require the **central moments**. These are derived algebraically from the raw moments:

1. **Mean** ($\mu$):
$$\mu = E[X]$$

2. **Variance** ($\sigma^2$):
$$\sigma^2 = E[X^2] - (E[X])^2$$

   To prevent division-by-zero errors in stable environments where variance is near-zero, we apply a numerical floor: $\sigma^2 = \max(\sigma^2, \epsilon)$ with $\epsilon = 10^{-8}$.

3. **Skewness** ($\gamma$): The third central moment $\mu_3 = E[(X - \mu)^3]$ is expanded as:
$$\mu_3 = E[X^3 - 3X^2\mu + 3X\mu^2 - \mu^3]$$
$$= E[X^3] - 3\mu E[X^2] + 2\mu^3 \qquad (13)$$

   The Fisher-Pearson coefficient of skewness is then:
$$\gamma = \frac{\mu_3}{\sigma^3} = \frac{E[X^3] - 3\mu E[X^2] + 2\mu^3}{(E[X^2] - \mu^2)^{3/2}}$$

## C. Prototype Refinement and Hybrid Fusion

This section details the update rules for the Hybrid VLM Teacher, specifically the explicit formulation for prototype updates and the handling of generative VLM embeddings. Prototype here works like a memory system in LLM (Zhu et al., 2026b; Xu et al., 2026a), which ensures the whole system works in an online manner.

### C.1. Online Prototype Update Rule

To bridge the domain gap, we dynamically refine the visual prototypes (class centroids) $\mathcal{C} = \{c_1, \ldots, c_K\}$. Let $c_k^{(t)}$ be the prototype for class $k$ at step $t$, and $N_k^{(t)}$ be the cumulative count of *verified clean* samples assigned to class $k$.

Upon receiving a batch of test samples, we first filter them using the current Adaptive Router. Only samples satisfying $\Delta > \tau$ are used for updates. For a batch of certified clean samples $\mathcal{B}_{clean}$ belonging to class $k$, the prototype is updated via Cumulative Moving Average (CMA) to ensure statistical inertia:

$$c_k^{(t)} = \text{Normalize} \left( \frac{N_k^{(t-1)} c_k^{(t-1)} + \sum_{x \in \mathcal{B}_{clean}^k} z_{img}(x)}{N_k^{(t-1)} + |\mathcal{B}_{clean}^k|} \right),$$
$$(14)$$

where $z_{img}(x)$ is the normalized visual embedding of the input. The Normalize($\cdot$) operation projects the vector back onto the unit hypersphere ($\|c\|_2 = 1$), ensuring compatibility with cosine similarity metrics. This explicit update

allows the VLM to gradually adapt its visual expectation from generic pre-training features to the specific distribution of the test stream.

### C.2. Hybrid Fusion Strategy

The VLM logit $S_{VLM}$ is computed as a weighted fusion:

$$S_{VLM} = \lambda \cdot S_{text} + (1 - \lambda) \cdot S_{proto} \qquad (15)$$

To ensure score compatibility between the two streams:

- **Embedding Models (CLIP/SigLIP):** Both $S_{text}$ and $S_{proto}$ represent cosine similarities, naturally bounded in $[-1, 1]$.

- **Generative Models (Qwen/LLaVA):** For $S_{text}$, we calculate the log-likelihood of the class name tokens. For $S_{proto}$, we extract the visual features $z_{img}$ directly from the underlying Vision Encoder (e.g., the ViT component before the LLM projector). This allows us to maintain a visual prototype space even for generative models. The text scores are min-max normalized within the batch to match the scale of cosine similarity before fusion.

## D. Experimental Setting Details

### D.1. Warm-up and Initialization Strategy

A critical challenge in online statistical defense is the "cold start" problem, where moment estimates (especially skewness) are unstable due to insufficient sample size. We implement a robust warm-up strategy:

1. **Confidence Weighting:** We define a confidence coefficient $\alpha_t = \min(1.0, \frac{N_t}{N_{warmup}})$, where $N_{warmup} \approx 100$.

2. **Correction Dampening:** During the warm-up phase, the skewness correction term in the Cornish-Fisher expansion is dampened by $\alpha_t$. The effective threshold becomes:

$$\tau_{eff} = \mu + \left[ \zeta + \alpha_t \cdot \frac{\gamma}{6}(\zeta^2 - 1) \right] \cdot \sigma \qquad (16)$$

   When $N_t$ is small ($\alpha_t \to 0$), the threshold reverts to a conservative Gaussian estimate ($\tau = \mu + \zeta\sigma$), preventing erratic behavior from noisy skewness estimates. As $N_t$ grows, the system smoothly transitions to full skewness-aware adaptive thresholding.

### D.2. Vision-Language Model Configuration

**Discriminative Backbones.** We utilize official pre-trained weights from OpenCLIP. Specifically, we employ `ViT-B-32` pre-trained on LAION-2B for CLIP and

*Table 6.* **Evaluation of ML Trust Chain on Medical Dataset (LC25000).** The Trust Chain strategy effectively bridges the gap between the low-accuracy trusted anchor (CLIP) and the high-accuracy untrusted expert (QuiltNet).

| Scenario | Configuration / Pipeline | Role | CA (%) | ASR (%) |
|---|---|---|---|---|
| *Baselines* | Trusted Universal VLM (CLIP) | Auditor | 53.7 | 0.6 |
| | Untrusted Domain VLM (QuiltNet) | Auditor | 75.2 | 98.1 |
| | Poisoned Victim Model | Victim | **98.7** | 98.3 |
| *Direct Auditing* | CLIP $\xrightarrow{audit}$ Victim | Defense | 87.9 | **3.5** |
| | Poisoned QuiltNet $\xrightarrow{audit}$ Victim | Defense | 96.4 | 98.4 (Fail) |
| *Trust Chain (Ours)* | Step 1: CLIP $\xrightarrow{audit}$ QuiltNet | Intermediate | 79.2 | 3.2 |
| | Step 2: **Sanitized QuiltNet** $\xrightarrow{audit}$ Victim | **Final Defense** | **95.8** | **4.5** |

`ViT-B-16` for SigLIP. Embeddings are normalized to the hypersphere.

**Generative Backbones.** For models like Qwen2.5-VL and LLaVA-1.5, we utilize a Key-Value (KV) cache strategy to optimize inference. We employ a hierarchical filtering strategy where a lightweight CLIP model first selects the Top-$K$ (default $K = 100$) candidates, which are then re-ranked by the Generative VLM using the prompt: *"Please answer with exactly one label from the list above."*

### D.3. PRISM Hyperparameters

**Hybrid Fusion Weight ($\lambda$).** We set $\lambda = 0.5$ by default (denoted as `text_anchor_weight`), balancing pre-trained semantic knowledge with online visual adaptation.

**Adaptive Thresholding.** We use a base Z-score coefficient $\zeta = -2.0$, corresponding to the lower 2.5% quantile of a normal distribution.

### D.4. Prompt Engineering

We maximize reproducibility by using standardized templates without extensive engineering. For numeric datasets (e.g., SVHN), class labels are mapped to natural language (e.g., "1" $\to$ "the digit one").

*Table 7.* Representative prompt templates used for VLM zero-shot anchors.

| Dataset | Template Structure |
|---|---|
| CIFAR-10/100 | `"a photo of a {label}."` |
| GTSRB | `"a close up photo of a '{label}' traffic sign."` |
| Food101 | `"a photo of {label}, a type of food."` |
| DTD | `"a photo of a {label} texture."` |
| Medical | `"a histopathology image of {label}."` |

## E. Case Study: Building an ML Trust Chain in Specialized Domains with a Backdoored Auditor

We investigate a "worst-case" scenario where the auditor itself is backdoored in medical image processing, an area that is highly sensitive to safety (Lin et al., 2026d; Li et al., 2025c). We use the LC25000 histopathological dataset. We

assume a **Collusive Threat Model** where both the Victim Model (ResNet) and the Specialized Auditor (QuiltNet) are poisoned with the same trigger.

**The Trust Chain Solution.** We construct a hierarchical pipeline: Trusted Weak Generalist (CLIP) → Untrusted Strong Specialist (QuiltNet) → Victim. First, the weak but clean Generalist (CLIP) audits the Untrusted Specialist. Although CLIP has low classification accuracy (53.7%), it effectively detects the distributional anomaly of the trigger, sanitizing the input stream for QuiltNet. The sanitized QuiltNet then audits the Victim. Results in Table 6 show this method recovers Clean Accuracy to 95.8% while suppressing ASR to 4.5%. This mechanism of progressive security insight draws inspiration from the "progressive diagnosis" (Zhu et al., 2026a) and mirrors multi-dimensional auditing strategies recently proposed for system-level intrusion detection (Yang et al., 2026a).

While PRISM demonstrates strong performance with trusted general-purpose VLMs, a practical dilemma arises in specialized domains like healthcare. General VLMs (e.g., CLIP) often lack the necessary domain expertise, while specialized VLMs (e.g., QuiltNet (Ikezogwo et al., 2023)) are typically third-party assets that may themselves be untrusted. In this section, we investigate a "worst-case" scenario where the auditor itself is backdoored. We propose a *Machine Learning Trust Chain* paradigm to secure the inference pipeline, using the LC25000 histopathological dataset as a testbed.

**Collusive Threat Model.** We assume a stringent threat environment involving two compromised models. First, the **Victim Model** (ResNet) is embedded with a standard backdoor. Second, **the Specialized Auditor (QuiltNet) is also poisoned**. Crucially, to maximize the difficulty of detection, we employ a *Collusive Threat Model*: we use the BadCLIP (Liang et al., 2024) framework to inject the **exact same trigger** into the QuiltNet auditor as used in the victim model. This creates a scenario where both the victim and the auditor are aligned to misclassify the trigger, theoretically bypassing standard consistency checks. The only "Root of Trust" is a standard, off-the-shelf **Generalist VLM** (CLIP),

which is assumed to be clean but possesses poor zero-shot accuracy ($\sim$53%) on this specialized medical task.

**The Dilemma of Direct Auditing.** Directly deploying either VLM as a standalone auditor fails to balance security and utility. If we employ the trusted Generalist (CLIP) to audit the victim directly, it successfully suppresses the Attack Success Rate (ASR) to 3.5% due to its statistical independence from the trigger. However, its lack of domain knowledge leads to a catastrophic drop in Clean Accuracy (CA) from 98.7% to 87.9%, deeming it unusable for medical diagnosis. Conversely, utilizing the Specialized Auditor (QuiltNet) maintains high clean accuracy (96.4%) but fails completely as a defense mechanism. Because QuiltNet contains the same backdoor as the victim, it "colludes" with the victim's prediction on poisoned samples, resulting in an ASR of 98.4%.

**Establishing the Trust Chain.** To resolve this, we construct a hierarchical auditing pipeline: *Trusted Weak Generalist* $\rightarrow$ *Untrusted Strong Specialist* $\rightarrow$ *Victim*. The process operates in two stages. First, the weak but clean Generalist (CLIP) audits the input stream of the Untrusted Specialist (QuiltNet). Although CLIP cannot classify the medical images accurately, it is highly effective at detecting the distributional anomaly of the trigger pattern via our statistical margin test, effectively "sanitizing" the QuiltNet inference stream. Second, this dynamically sanitized QuiltNet is used to audit the final Victim Model.

**Results and Implications.** This hierarchical strategy effectively decouples the defense capability from the auditor's own integrity regarding the trigger. By filtering the specialized auditor's inputs using a generalist root of trust, we recover the system's security. The Trust Chain suppresses the ASR to 4.5%—comparable to using the clean CLIP directly—while achieving a final Clean Accuracy of 95.8%, significantly outperforming the naive generalist approach. This result confirms that PRISM can defend against a compromised victim model even when the domain-specific auditor contains the same backdoor, provided a clean (even if weak) foundation model is available to initiate the chain of trust.

## F. Empirical Validation of the Moment-Based Distribution Approximation

The core mechanism of PRISM's Adaptive Router relies on the observation that the logit margin $\Delta$—defined as the exponential difference between the VLM's support for the victim's prediction and the next best class—**exhibits significant asymmetry**. This distributional characteristic necessitates the use of the Cornish-Fisher expansion to calibrate thresholds, rather than relying on standard Gaussian statistics which assume symmetry. To validate this, we visualize

empirical distributions of $\Delta$ across diverse conditions.

The visualizations reveal that the distribution of benign samples (highlighted in blue) consistently deviates from normality. In the upper panel (Figure 8), which details the distribution under BadNets attacks across five distinct datasets, we observe that the distribution shape varies significantly by domain. For instance, simpler datasets like MNIST exhibit sharper peaks with heavy tails, whereas complex domains like ImageNet show broader variance. A standard normal approximation would fail to capture these tail behaviors, leading to either high false rejection rates or missed detections.

However, the **approximated density curves** (shown in yellow), derived via the **Edgeworth expansion** using the online estimates of the first three moments, demonstrate a tight alignment with the empirical histograms. This confirms that incorporating skewness $\gamma$ into the threshold calculation allows the decision boundary (red dashed line) to dynamically adapt to the natural asymmetry of the semantic margin without enforcing a strict parametric assumption.

Furthermore, the lower panel (Figure 10) demonstrates the robustness of this approximation across 12 different attack modalities on CIFAR-10. Despite the variation in attack strategies—ranging from patch-based BadNets to clean-image FLIP attacks—the benign distributions remain consistent, and the moment-based fitted curves accurately encapsulate the safe acceptance regions. Notably, the backdoored samples (highlighted in red) and misclassified inputs (highlighted in orange) predominantly fall into the lower-margin regions below the adaptive threshold. The separation illustrates that while the benign data manifold is complex and asymmetric, it is statistically distinct from the backdoor distribution, and our moment-based method provides a mathematically grounded approach to isolate it efficiently.

## G. Details of Adaptive Auditor-Aware Attack

To rigorously evaluate the robustness of PRISM under a strong adaptive threat model—where the adversary specifically targets the "trusted" VLM auditor—we design a *Semantic Typography Attack*. Unlike standard backdoor triggers that rely on abstract patterns, this attack embeds semantic text information into the image to mislead the VLM's zero-shot prediction.

**Implementation Details.** We conduct this experiment on the CIFAR-10 dataset targeting the "Airplane" class (Class 0). The attack inserts the text string "Plane" into the benign images. The text is rendered in a bright red color (RGB: 255, 0, 0). To minimize the occlusion of the original object's visual features, the text is placed in the **bottom-left corner** of the image with a 1-pixel margin. The text height

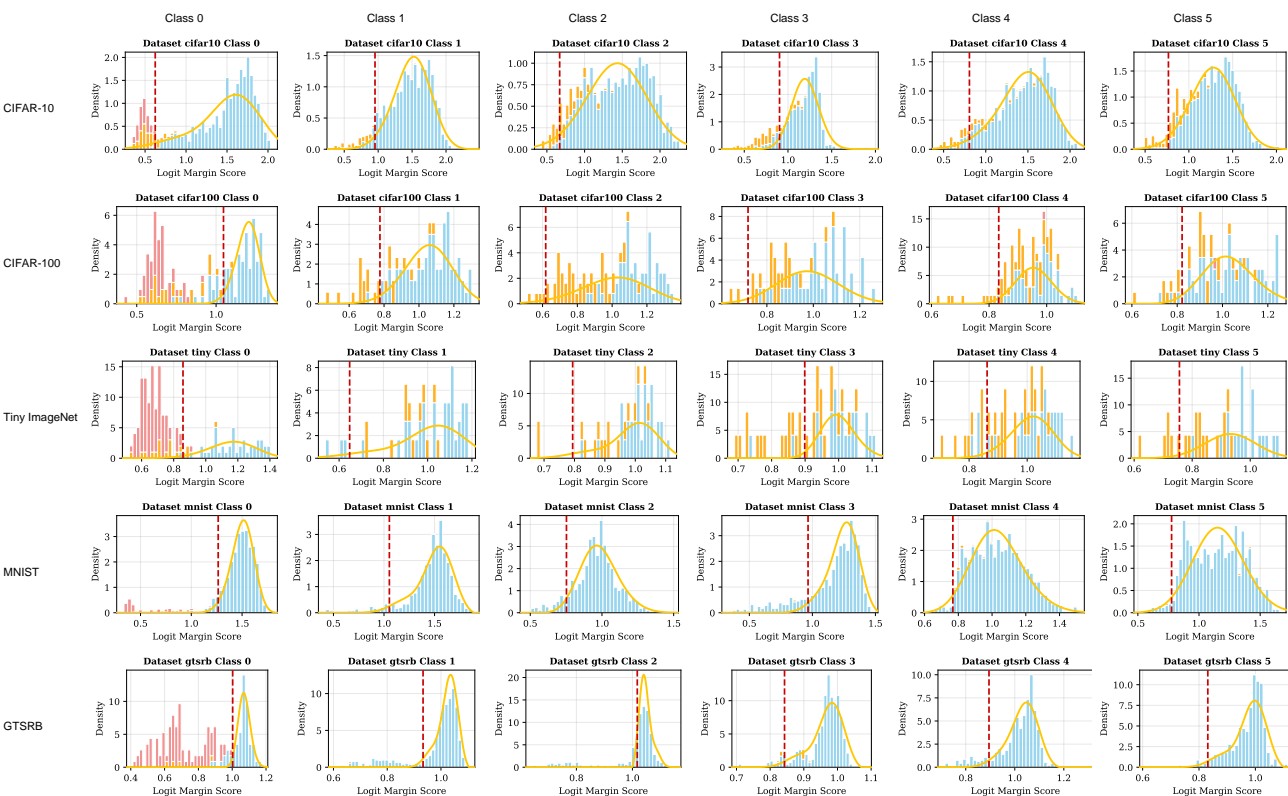

*Figure 8.* **Distribution of Logit Margin across different datasets (BadNets).** The histograms visualize the density of the logit margin $\Delta$. Benign samples are highlighted in `blue`, successful backdoor samples in `red`, and misclassified samples in `orange`. The yellow curve represents the **density approximation derived via Edgeworth expansion** using online moment estimates. The vertical red dashed line indicates the adaptive threshold $\tau$. The tight alignment validates that our skewness-aware modeling effectively distinguishes high-consistency benign samples from low-consistency attacks.

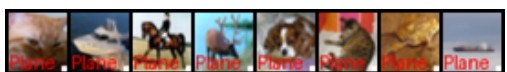

*Figure 9.* **Visualization of Adaptive Text Prompt Attack Samples.** The text "`Plane`" is superimposed on the bottom-left corner of the images (e.g., a Frog or a Cat) to act as a semantic trigger targeting the VLM auditor.

is controlled to be approximately $21\%$ of the image height.

Visualizations of the attack samples are provided in Figure 9. As observed, the red text trigger is distinct yet leaves the main body of the original object (e.g., the cat) visible, creating a scenario where the victim model (activated by the trigger) and the visual perception of the VLM (observing the object) may conflict.

## H. Defense against Feature Mixing Backdoors

To comprehensively evaluate the robustness of PRISM, we further consider the *Feature Mixing Backdoor* (FMB) attacks (Lin et al., 2020). Unlike standard patch-based attacks, FMB is designed to confuse VLM semantics by linearly blending features from multiple classes (e.g., mixing the pixel values of a source image with another image). We evaluate our defense against both the one-to-one (o2o) and all-to-one (a2o) attack variants.

The results are presented in Table 9. PRISM exhibits strong defense performance, suppressing the Attack Success Rate (ASR) to $\leq 2.2\%$ in both settings while maintaining or even slightly improving Clean Accuracy (CA).

**Analysis.** The effectiveness of PRISM against FMB can be attributed to the semantic inconsistency it introduces. While feature mixing may successfully confuse the VLM between the two mixed *source* classes (e.g., the VLM might hesitate between "Cat" and "Dog"), the resulting image does not semantically resemble the malicious *target* label (e.g., "Airplane"). Since the victim model is forced to predict the target label with high confidence, while the VLM finds no semantic evidence for that target, the logit margin used by PRISM remains high. This allows our adaptive router to correctly flag these inputs as anomalous distributions.

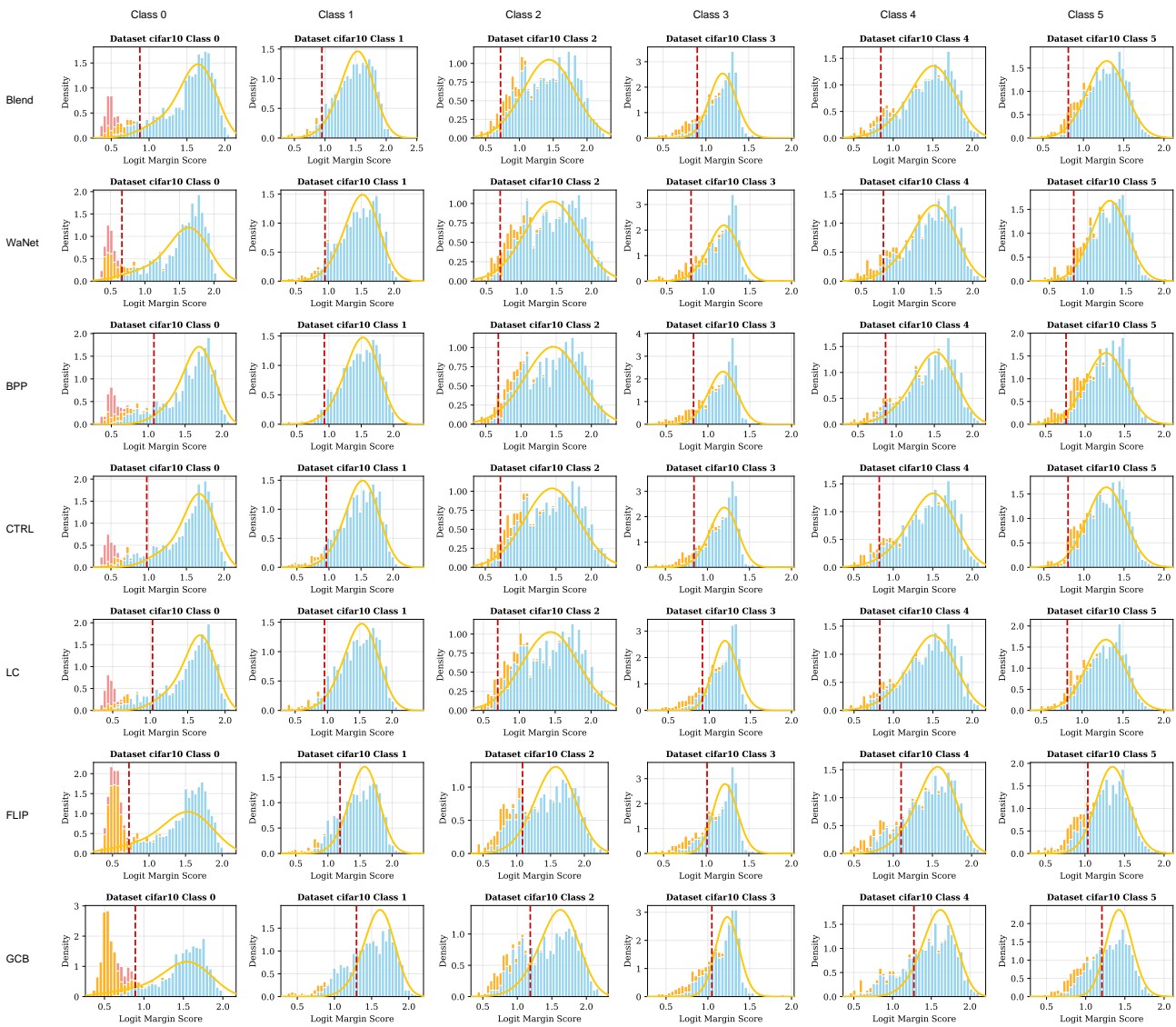

*Figure 10.* **Distribution of Logit Margin across different attack types (CIFAR-10).** This visualization validates the robustness of our **moment-based approximation** against 12 diverse backdoor attacks, including dynamic and clean-label variants. Despite the diversity in attack vectors, the benign distributions maintain characteristic asymmetries that are accurately captured by our adaptive thresholding logic.

## I. Performance Comparison on Domain-Specific Datasets

Table 8 presents a detailed comparison on the GTSRB dataset under a stealthy 1% poison rate. This scenario is particularly challenging for existing defenses: model repairing methods like CLP and C&C often damage the Clean Accuracy (CA) significantly due to the domain gap, while input robustness methods fail to detect the sparse trigger samples. As shown by the extensive red highlights, all baseline methods fail to simultaneously maintain high CA and low ASR. In contrast, PRISM successfully suppresses the average ASR to 3.6% while maintaining a high CA of 93.4%, outperforming the best baseline by a significant margin. This underscores PRISM's superiority in specialized domains where pre-trained knowledge must be carefully adapted.

## J. Defense Independence from Zero-Shot Performance

A common concern is whether VLM-based defenses rely heavily on the VLM's inherent zero-shot classification accuracy. Table 10 addresses this by evaluating PRISM across representative datasets with varying levels of zero-shot performance, ranging from 62.3% (CIFAR-100) down to 25.1% (GTSRB). To rigorously quantify detection efficacy, we adopt a utility-centric metric formulation: **True Positive Rate (TPR)** is calculated strictly over *effective* attacks (backdoor samples that successfully fool the victim model into the target class), while **False Positive Rate (FPR)** is com-

*Table 8.* Evaluation of test-time backdoor defenses on GTSRB (1% poison rate). ASRs below 15% are highlighted in blue to indicate a successful defense, while ASRs above 15% are denoted in red as failed defenses.

| Defense → | | No Defense | | CLP | | ScaleUp | | BDMAE | | ZIP | | TeCo | | C&C | | LPP | | Refine | | Zero-shot | | Ensemble | | PRISM (Ours) | |
|---|---|---|---|---|---|---|---|---|---|---|---|---|---|---|---|---|---|---|---|---|---|---|---|---|---|
| Attack ↓ | | CA | ASR | CA | ASR | CA | ASR | CA | ASR | CA | ASR | CA | ASR | CA | ASR | CA | ASR | CA | ASR | CA | ASR | CA | ASR | CA | ASR |
| Classic | BadNet | 98.5 | 91.4 | 97.1 | 16.7 | 92.6 | 0.1 | 97.7 | 5.1 | 98.6 | 1.5 | 98.5 | 91.3 | 88.0 | 5.1 | 96.8 | 0.4 | 97.9 | 0.1 | 25.1 | 0.3 | 91.7 | 45.2 | 95.6 | 0.0 |
| | Blend | 98.4 | 93.4 | 95.1 | 16.1 | 98.3 | 43.9 | 97.4 | 42.6 | 97.2 | 9.9 | 98.4 | 93.4 | 86.8 | 23.0 | 98.4 | 42.4 | 98.3 | 0.1 | 25.1 | 0.1 | 90.3 | 26.0 | 93.6 | 2.5 |
| Dynamic | WaNet | 98.6 | 70.7 | 89.8 | 47.8 | 97.5 | 31.1 | 96.4 | 25.5 | 93.4 | 49.3 | 98.6 | 70.4 | 95.4 | 49.5 | 93.5 | 11.3 | 97.4 | 1.3 | 25.1 | 0.2 | 92.4 | 16.0 | 93.0 | 0.8 |
| | BPP | 98.4 | 75.3 | 97.6 | 0.2 | 98.3 | 35.2 | 97.7 | 38.1 | 97.5 | 7.4 | 98.8 | 75.2 | 95.3 | 2.0 | 98.2 | 0.0 | 97.7 | 0.1 | 25.1 | 0.0 | 96.2 | 29.5 | 95.7 | 0.0 |
| | IAB | 98.3 | 67.3 | 94.8 | 1.2 | 98.4 | 39.9 | 97.1 | 17.5 | 97.4 | 12.7 | 93.0 | 31.3 | 88.9 | 15.2 | 97.3 | 3.4 | 98.4 | 0.0 | 25.1 | 0.1 | 91.9 | 5.1 | 95.7 | 0.0 |
| | SSBA | 98.3 | 95.8 | 94.8 | 62.8 | 98.3 | 92.0 | 97.3 | 86.9 | 96.6 | 64.5 | 91.9 | 6.9 | 79.5 | 88.6 | 94.5 | 68.5 | 98.3 | 0.8 | 25.1 | 0.1 | 91.1 | 45.1 | 90.9 | 7.6 |
| Clean Label | CTRL | 98.4 | 79.5 | 95.2 | 6.0 | 98.3 | 73.4 | 98.3 | 36.7 | 96.8 | 0.2 | 93.1 | 14.7 | 60.0 | 1.5 | 93.2 | 0.0 | 98.3 | 0.2 | 25.1 | 0.0 | 89.9 | 39.3 | 92.7 | 6.7 |
| | SIG | 98.4 | 78.2 | 87.9 | 45.7 | 98.4 | 41.9 | 97.2 | 46.0 | 97.1 | 43.5 | 98.4 | 67.8 | 78.2 | 16.9 | 97.8 | 40.4 | 98.1 | 0.5 | 25.1 | 0.0 | 90.3 | 31.6 | 92.7 | 0.8 |
| | LC | 98.4 | 80.5 | 95.2 | 1.5 | 98.4 | 69.9 | 97.6 | 3.8 | 97.1 | 0.1 | 93.4 | 21.7 | 86.5 | 0.6 | 98.3 | 1.0 | 98.3 | 0.0 | 25.1 | 0.2 | 90.8 | 31.0 | 93.0 | 9.9 |
| Clean Image | FLIP | 95.6 | 38.8 | 86.3 | 0.7 | 94.5 | 3.4 | 91.2 | 4.6 | 91.0 | 3.5 | 95.6 | 38.7 | 64.4 | 9.5 | 95.0 | 2.2 | 94.6 | 25.1 | 25.1 | 0.0 | 88.2 | 0.9 | 92.2 | 4.8 |
| | GCB | 97.3 | 98.6 | 94.8 | 90.9 | 97.2 | 84.3 | 96.4 | 87.2 | 96.0 | 84.8 | 87.7 | 14.3 | 77.1 | 98.8 | 89.3 | 55.1 | 96.9 | 93.0 | 25.1 | 0.5 | 90.9 | 46.9 | 91.9 | 6.7 |
| Average | | 98.0 | 79.0 | 93.5 | 26.3 | 97.3 | 46.8 | 96.8 | 35.8 | 96.2 | 25.2 | 95.2 | 47.8 | 81.8 | 28.2 | 95.7 | 20.4 | 97.7 | 11.0 | 25.1 | 0.1 | 91.2 | 28.8 | 93.4 | 3.6 |
| CA Drop (smaller is better) | | | | ▼4.5 | | ▼0.7 | | ▼1.3 | | ▼1.8 | | ▼2.8 | | ▼16.2 | | ▼2.4 | | ▼0.4 | | ▼72.9 | | ▼6.8 | | ▼4.7 | |
| ASR Drop (larger is better) | | | | | ▼52.7 | | ▼32.2 | | ▼43.2 | | ▼53.8 | | ▼31.3 | | ▼50.8 | | ▼58.6 | | ▼68.0 | | ▼78.9 | | ▼50.3 | | ▼75.4 |

*Table 9.* **Defense performance against Feature Mixing Backdoors (FMB).** We report Clean Accuracy (CA) and Attack Success Rate (ASR) percentages.

| FMB Variant | No Defense | | PRISM | |
|---|---|---|---|---|
| | CA | ASR | CA | ASR |
| One-to-One | 92.0 | 85.0 | 93.4 | 0.0 |
| All-to-One | 91.5 | 77.7 | 92.9 | 2.2 |

puted solely on *correctly classified* clean samples. This unique calculation excludes ineffective attacks and already-misclassified clean inputs, preventing them from artificially inflating detection performance. Remarkably, even on GTSRB where the VLM's raw accuracy is nearly random, PRISM effectively utilizes *relative* semantic inconsistency to distinguish attacks, achieving a high average TPR of 96.3% while maintaining a low FPR of 6.8%. The results confirm that our framework successfully recovers the victim model's high utility (raising CA from the VLM's 25.1% to 93.4%) while consistently suppressing ASR, validating that PRISM operates as a robust semantic auditor rather than a simple classifier substitute.

## K. Robustness to Varying Poison Rates

Table 11 examines the sensitivity of PRISM to the adversary's injection capability, testing poison rates of 1%, 5%, and 10% on both CIFAR-10 and GTSRB. Statistical defenses often struggle at extremes: low poison rates yield insufficient samples for estimation, while high rates can dominate the clean distribution. PRISM demonstrates consistent robustness across this spectrum. Even at a 10% poison rate on GTSRB, where the attack distribution is strong, PRISM maintains an ASR of 4.0% with negligible impact on clean accuracy. This stability is attributed to our online warm-up and selective update mechanism, which prevents the statistical moments from being skewed by the poisoned samples.

## L. Scalability Across 17 Datasets and VLM Architectures

To demonstrate the universality of our framework, Table 12 provides an extensive evaluation across 17 diverse datasets, comparing the standard CLIP backbone with the generative Qwen2.5-VL-7b. The results confirm that PRISM generalizes well to datasets with extreme class imbalances (e.g., Caltech101, IR=86.5) and varying granularities (e.g., ImageNet, 1000 classes). Notably, the generative Qwen2.5-VL-7b backbone achieves comparable or superior performance to CLIP, particularly in fine-grained tasks like StanfordCars and Food101. This indicates that PRISM is model-agnostic and can benefit from the continuous advancements in Large Vision-Language Models.

## M. Component Contribution Analysis

Table 13 offers a granular ablation study dissecting the contribution of Online Update, Prototype Refinement, and Skewness Correction. The contrast between CIFAR-100 (general domain) and GTSRB (specialized domain) is revealing.

- **Necessity of Online Update:** On GTSRB, removing the online update mechanism leads to a catastrophic failure (ASR surges to 23.7%), as the static VLM anchors fail to align with the traffic sign distribution.

- **Impact of Prototype Refinement:** Without visual prototypes, the Clean Accuracy drops significantly (e.g., -13.6% on GTSRB), highlighting their role in bridging the domain gap.

- **Role of Skewness Correction:** Removing skewness correction destabilizes the thresholding, causing increased false acceptances of attacks (higher ASR) or false rejections of clean data (lower CA).

*Table 10.* Results of PRISM on five representative datasets. The table includes Clean Accuracy (CA), Attack Success Rate (ASR), True Positive Rate (TPR), and False Positive Rate (FPR). PRISM defends against all 11 attacks across all datasets.

| Dataset→ | CIFAR10 (86.7%) | | | | CIFAR100 (62.3%) | | | | Tiny (57.8%) | | | | MNIST (43.6%) | | | | GTSRB (25.1%) | | | |
|---|---|---|---|---|---|---|---|---|---|---|---|---|---|---|---|---|---|---|---|---|
| Attack↓ | CA | ASR | TPR | FPR | CA | ASR | TPR | FPR | CA | ASR | TPR | FPR | CA | ASR | TPR | FPR | CA | ASR | TPR | FPR |
| BadNet | 93.9 | 0.0 | 98.9 | 3.4 | 71.5 ▲1.3 | 0.0 | 100.0 | 9.3 | 62.9 ▲6.4 | 0.0 | 99.0 | 12.5 | 99.4 ▼0.1 | 7.8 | 98.9 | 5.5 | 95.6 ▼2.8 | 0.0 | 99.0 | 2.9 |
| Blend | 94.2 | 0.0 | 100.0 | 4.7 | 72.4 ▲1.5 | 0.0 | 100.0 | 9.8 | 58.7 ▲1.9 | 0.0 | 99.0 | 9.8 | 99.3 ▼0.1 | 6.7 | 100.0 | 5.5 | 93.6 ▼4.9 | 2.5 | 95.7 | 4.9 |
| WaNet | 93.6 | 0.0 | 98.8 | 3.2 | 67.4 ▲3.7 | 0.0 | 100.0 | 10.6 | 57.5 ▲1.5 | 3.0 | 95.0 | 8.2 | 93.7 ▼5.1 | 7.4 | 93.3 | 4.9 | 93.0 ▼5.6 | 0.8 | 93.9 | 12.3 |
| BPP | 93.7 | 1.1 | 100.0 | 3.0 | 69.5 ▲4.7 | 0.0 | 100.0 | 11.3 | 56.9 ▼0.9 | 0.0 | 96.3 | 9.0 | 99.3 ▼0.1 | 3.3 | 95.0 | 5.5 | 95.7 ▼0.7 | 0.0 | 100.0 | 1.9 |
| IAB | 92.4 | 0.0 | 100.0 | 4.2 | 69.8 ▲4.8 | 0.0 | 100.0 | 10.0 | 57.8 ▲1.9 | 0.0 | 98.8 | 11.2 | 93.1 ▼3.9 | 8.4 | 95.6 | 4.9 | 95.7 ▼2.6 | 0.0 | 100.0 | 3.2 |
| SSBA | 93.8 | 0.0 | 98.8 | 4.1 | 72.2 ▲2.0 | 0.0 | 97.8 | 8.5 | 62.7 ▲6.0 | 0.0 | 100.0 | 19.1 | 94.5 ▼2.4 | 2.4 | 95.1 | 3.2 | 90.9 ▼7.4 | 7.6 | 93.8 | 10.7 |
| CTRL | 94.2 | 0.0 | 100.0 | 3.1 | 72.3 ▲2.0 | 0.0 | 100.0 | 11.2 | 55.1 ▼1.9 | 0.0 | 100.0 | 10.7 | 99.4 ▼0.3 | 0.0 | 10.0 | 0.1 | 92.7 ▼5.9 | 6.7 | 91.9 | 6.8 |
| SIG | 92.6 | 2.2 | 98.8 | 4.5 | 72.0 ▲1.8 | 0.0 | 100.0 | 6.6 | 62.3 ▲5.8 | 0.0 | 100.0 | 10.3 | 99.2 ▼0.1 | 0.0 | 100.0 | 0.1 | 92.7 ▼5.6 | 0.8 | 98.1 | 6.0 |
| LC | 94.6 | 0.0 | 100.0 | 2.2 | 73.2 ▲1.9 | 0.0 | 100.0 | 9.4 | 62.5 ▲6.3 | 0.0 | 100.0 | 11.6 | 99.2 ▼0.1 | 0.0 | 100.0 | 0.0 | 93.0 ▼5.4 | 9.9 | 94.0 | 12.1 |
| FLIP | 91.7 | 1.1 | 100.0 | 12.2 | 70.0 ▼0.1 | 1.0 | 100.0 | 11.7 | 56.4 ▼0.1 | 0.0 | 92.7 | 9.6 | 96.2 ▼1.1 | 6.7 | 97.4 | 1.2 | 92.2 ▼3.4 | 4.8 | 100.0 | 0.4 |
| GCB | 90.1 | 4.4 | 99.1 | 19.6 | 68.1 ▼2.1 | 2.2 | 92.6 | 21.1 | 62.4 ▲5.5 | 2.0 | 97.5 | 39.3 | 94.7 ▼1.3 | 3.3 | 90.5 | 10.0 | 91.9 ▼5.3 | 6.7 | 92.7 | 13.2 |
| Average | 93.2 | 0.8 | 99.5 | 5.8 | 70.8 ▲2.0 | 0.3 | 99.1 | 10.9 | 59.9 ▲3.3 | 0.5 | 98.0 | 13.8 | 97.3 ▼1.2 | 4.4 | 88.7 | 3.7 | 93.4 ▼4.5 | 3.6 | 96.3 | 6.8 |

These results confirm that all three components are essential for a robust, domain-adaptive defense.

## N. Sensitivity to Target Class Selection

In the main evaluation, we set the target label $y_t = 0$ following BackdoorBench (Wu et al., 2022). Class 0 (Speed 20 km/h) is actually the *minority class* in GTSRB at only 0.54% of training data, representing a conservative choice. We ablate over 5 target classes spanning a $10.7\times$ frequency range, evaluated under 9 attacks (FLIP and GCB excluded as they rely on pre-generated files that cannot be re-run with an arbitrary target).

Table 14 shows that ASR ranges from 0.38% to 3.14% with no statistically significant monotonic trend across a $10.7\times$ frequency range. Variation reflects specific class semantics, not frequency. PRISM's detection signal—the logit margin between the VLM's support for the victim's prediction and the next best class—is class-frequency-agnostic by design.

## O. Robustness Across Victim Architectures

The main experiments use PreActResNet18 as the victim architecture. We extend the evaluation to MobileNetV2, VGG-16, and ViT-B/32 on CIFAR-10, reporting per-attack-category averaged CA/ASR across all attacks in each category. Pre-defense ASRs ranged from 67.8% to 99.7% across all architectures and attack types.

PRISM reduces all per-category ASRs to under 2% across every architecture and attack category (Table 15). This architecture-agnosticism is structural: PRISM wraps the victim model externally and never accesses its internal parameters. The defense signal is computed entirely within the VLM branch, so effectiveness depends on the VLM's feature space rather than the victim's architecture. This implies that PRISM remains compatible with emerging neural patterns, such as vision state space models (Ke et al., 2026).

## P. Dirty Cold-Start Robustness

The online prototype initialization is vulnerable if the first batch of test samples is entirely poisoned (a *dirty cold-start*), as the centroids may be initialized with trigger-contaminated features. PRISM addresses this through its **text anchor** mechanism: since text anchors are derived from the frozen VLM's linguistic knowledge and require no visual samples, they provide a statistically safe initialization that remains active until the prototype-based term stabilizes.

To stress-test this, we run the test stream with poison injected from the very beginning (including cold-start initialization). Poison rate here is defined as the fraction of the test stream that consists of poisoned target-class samples. We compare Clean Warmup (clean initialization batch) vs. Dirty Cold-Start (initialization batch 100% poisoned) across 4 attacks and 3 poison rates on CIFAR-10.

Table 16 confirms that dirty cold-start cannot compromise PRISM. Before any prototype exists, text anchors dominate: they are derived from the frozen VLM's linguistic knowledge and do not support the backdoor-target prediction, so triggered samples fail the gate and cannot corrupt statistics or prototypes. PRISM remains text-anchor-only until sufficient clean samples arrive; the warm-up phase (App. D.1) accelerates prototype convergence for performance, not security. Even at 50% poison rate, the worst-case ASR is 10.9% (Blend), versus ∼95% undefended, with a gap of ≤5% relative to clean warmup across all conditions.

## Q. Comparison with Training-Time Defenses

We provide this comparison for contextualization. All listed baselines except PRISM require full training data access; PRISM operates data-free at test time. Results show ASR (%) on CIFAR-10 at 5% poison rate; Avg CA is in the last row. Training-time baselines use BackdoorBench (Wu et al., 2022) implementations.

*Table 11.* PRISM under poison rate of 1%, 5%, 10%. Results are shown with change values on the right side. PRISM can succeed under all poison rates and all datasets.

| Dataset→ | CIFAR10 | | | | | | GTSRB | | | | | |
|---|---|---|---|---|---|---|---|---|---|---|---|---|
| Poison Rate→ | 1% | | 5% | | 10% | | 1% | | 5% | | 10% | |
| Attack↓ | CA | ASR | CA | ASR | CA | ASR | CA | ASR | CA | ASR | CA | ASR |
| BadNets | 94.4 ▲1.2 | 0.0 ▼73.8 | 93.9 ▲1.6 | 0.0 ▼87.9 | 94.0 ▲2.2 | 0.0 ▼93.8 | 95.6 ▼2.8 | 0.0 ▼91.4 | 96.5 ▼0.8 | 0.8 ▼92.7 | 96.2 ▲0.2 | 0.5 ▼94.1 |
| Blend | 94.5 ▲0.7 | 0.0 ▼94.1 | 94.2 ▲0.7 | 0.0 ▼99.4 | 94.0 ▲0.3 | 0.0 ▼99.8 | 93.6 ▼4.9 | 2.5 ▼97.4 | 95.9 ▼2.7 | 0.8 ▼99.1 | 95.1 ▲1.8 | 1.0 ▼99.0 |
| WaNet | 93.7 ▲2.5 | 0.0 ▼72.0 | 93.6 ▲2.5 | 0.0 ▼94.1 | 93.3 ▲2.7 | 0.0 ▼96.9 | 93.0 ▼5.6 | 0.8 ▼69.9 | 92.1 ▼3.6 | 9.2 ▼89.5 | 91.5 ▼2.6 | 12.0 ▼85.6 |
| BPP | 93.4 ▲2.0 | 1.1 ▼98.1 | 93.7 ▲2.1 | 1.1 ▼98.3 | 93.4 ▲2.0 | 1.1 ▼98.1 | 95.7 ▼0.7 | 0.0 ▼99.8 | 93.7 ▼2.7 | 0.0 ▼99.8 | 93.3 ▲3.9 | 0.0 ▼98.8 |
| IAB | 93.1 ▲2.6 | 0.0 ▼54.6 | 92.4 ▲1.0 | 0.0 ▼95.0 | 92.8 ▲3.1 | 0.0 ▼94.9 | 95.7 ▼2.6 | 0.0 ▼67.3 | 94.4 ▼3.1 | 0.0 ▼83.2 | 93.5 ▼3.7 | 0.0 ▼92.4 |
| SSBA | 94.0 ▲0.6 | 0.0 ▼99.7 | 93.8 ▲0.8 | 0.0 ▼97.3 | 94.1 ▲1.2 | 0.0 ▼97.3 | 90.9 ▼7.4 | 7.6 ▼88.2 | 96.8 ▼1.4 | 0.0 ▼99.3 | 96.6 ▲0.2 | 0.5 ▼98.5 |
| CTRL | 94.7 ▲0.7 | 0.0 ▼55.3 | 94.2 ▲0.6 | 0.0 ▼95.9 | 94.2 ▲0.6 | 0.0 ▼95.9 | 92.7 ▼5.9 | 6.7 ▼88.6 | 91.0 ▼7.6 | 7.6 ▼87.7 | 90.5 ▼5.7 | 8.5 ▼90.4 |
| SIG | 92.6 ▼1.1 | 3.3 ▼77.0 | 92.6 ▼1.0 | 2.2 ▼91.6 | 92.4 ▲7.8 | 1.7 ▼96.3 | 92.7 ▼5.6 | 0.8 ▼77.3 | 91.7 ▼6.5 | 1.7 ▼88.8 | 91.8 ▲4.3 | 2.0 ▼93.9 |
| LC | 94.5 ▲1.0 | 0.0 ▼68.3 | 94.6 ▲1.1 | 0.0 ▼98.4 | 93.7 ▲9.3 | 1.1 ▼98.7 | 93.0 ▼5.4 | 9.9 ▼70.6 | 92.1 ▼6.3 | 10.9 ▼59.9 | 91.5 ▼5.1 | 11.5 ▼78.8 |
| FLIP | 93.9 ▲0.6 | 0.0 ▼98.1 | 91.7 ▲1.9 | 1.1 ▼98.1 | 92.6 ▲7.2 | 2.2 ▼97.5 | 92.2 ▼3.4 | 4.8 ▼34.0 | 91.9 ▼1.8 | 2.0 ▼54.4 | 91.4 ▼0.7 | 2.5 ▼81.2 |
| GCB | 92.5 ▲0.0 | 3.3 ▼96.7 | 90.1 ▲1.6 | 4.4 ▼95.6 | 90.0 ▲5.8 | 5.6 ▼94.4 | 91.9 ▼5.3 | 6.7 ▼92.0 | 88.8 ▼6.5 | 5.0 ▼94.9 | 87.7 ▼5.8 | 6.0 ▼94.0 |
| Average | 93.8 ▲1.0 | 0.7 ▼80.7 | 93.2 ▲1.2 | 0.8 ▼95.6 | 93.1 ▲3.8 | 1.1 ▼96.7 | 93.4 ▼4.5 | 3.6 ▼79.7 | 93.2 ▼3.9 | 3.5 ▼86.3 | 92.6 ▼3.0 | 4.0 ▼91.5 |

*Table 12.* Expanded scalability defense results comparing CLIP and Qwen2.5-VL-7b under poison rate of 5%. ASRs below 20% are highlighted in blue (success), while ASRs above 20% are denoted in red (fail). **Qwen2.5-VL-7b** demonstrates robust defense performance comparable to or better than CLIP across most datasets.

| Dataset | IR | Entropy | # Class | Size | RN50 | CLIP | | | | | | | Qwen2.5-VL-7b | | | | | | |
|---|---|---|---|---|---|---|---|---|---|---|---|---|---|---|---|---|---|---|---|
| | | | | | | VLM | Blend | | BPP | | CTRL | | VLM | Blend | | BPP | | CTRL | |
| | | | | | | | ΔCA | ASR | ΔCA | ASR | ΔCA | ASR | | ΔCA | ASR | ΔCA | ASR | ΔCA | ASR |
| SVHN | 3.20 | 0.965 | 10 | 234M | 95.5 | 13.4 | ▼1.6 | 0.8 | ▼1.9 | 1.6 | ▼1.8 | 1.8 | 64.5 | ▼4.8 | 3.6 | ▼4.6 | 6.5 | ▼4.5 | 8.0 |
| Country211 | 1.00 | 1.00 | 211 | 20.9G | 6.8 | 17.2 | ▲5.3 | 0.0 | ▲5.4 | 1.4 | ▲5.3 | 0.0 | 21.7 | ▲10.1 | 0.0 | ▲10.2 | 0.0 | ▲10.6 | 0.0 |
| GTSRB | 12.5 | 0.920 | 43 | 689M | 97.8 | 32.6 | ▼4.9 | 2.5 | ▼0.7 | 0.0 | ▼5.9 | 6.7 | 52.5 | ▼4.8 | 0.8 | ▼5.9 | 6.4 | ▼4.7 | 0.8 |
| FER2013 | 16.0 | 0.932 | 7 | 950M | 63.0 | 41.4 | ▼5.6 | 0.0 | ▼3.8 | 4.8 | ▼4.7 | 4.8 | 27.6 | ▼4.4 | 4.8 | ▼2.6 | 4.8 | ▼1.2 | 3.2 |
| DTD | 1.00 | 1.00 | 47 | 127M | 25.7 | 44.3 | ▲17.8 | 0.0 | ▲16.3 | 0.0 | ▲19.6 | 0.0 | 60.3 | ▲31.8 | 0.0 | ▲27.3 | 0.0 | ▲28.4 | 0.0 |
| TinyImgNet | 1.00 | 1.00 | 200 | 1.2G | 57.2 | 57.8 | ▲1.9 | 0.0 | ▼0.9 | 0.0 | ▼1.8 | 0.0 | 57.6 | ▲4.4 | 0.0 | ▲2.5 | 0.0 | ▲4.7 | 0.0 |
| StanfordCars | 2.83 | 0.999 | 196 | 481M | 49.3 | 59.6 | ▲9.9 | 0.0 | ▲13.8 | 2.5 | ▲5.4 | 0.0 | 77.1 | ▲22.4 | 0.0 | ▲28.8 | 0.0 | ▲18.2 | 0.0 |
| ImageNet | 1.00 | 1.00 | 1000 | 3.7G | 66.1 | 62.4 | ▲5.8 | 6.2 | ▲7.0 | 5.0 | ▲5.4 | 3.1 | 51.0 | ▼2.5 | 0.0 | ▼2.6 | 0.0 | ▼3.4 | 0.0 |
| SUN397 | 37.7 | 0.953 | 397 | 146G | 55.3 | 62.5 | ▲12.9 | 0.9 | ▲14.2 | 0.9 | ▲13.0 | 0.9 | 45.2 | ▼4.2 | 7.3 | ▼2.4 | 7.3 | ▼0.2 | 3.9 |
| MNIST | 1.00 | 1.00 | 10 | 338M | 99.5 | 62.7 | ▲6.4 | 5.8 | ▲7.3 | 4.7 | ▲6.2 | 2.9 | 92.1 | ▼0.7 | 4.4 | ▼0.5 | 0.0 | ▼0.8 | 0.0 |
| CIFAR100 | 1.00 | 1.00 | 100 | 73.6G | 69.2 | 64.2 | ▲2.2 | 0.0 | ▲3.4 | 0.0 | ▲1.0 | 0.0 | 67.6 | ▲0.3 | 0.0 | ▲4.6 | 0.0 | ▼1.0 | 0.0 |
| Flowers102 | 1.00 | 1.00 | 102 | 676M | 31.0 | 66.5 | ▲15.2 | 0.0 | ▲12.7 | 0.0 | ▲17.7 | 0.0 | 78.0 | ▼2.4 | 10.0 | ▲2.6 | 0.0 | ▲0.7 | 0.0 |
| Caltech101 | 86.5 | 0.902 | 101 | 324M | 68.8 | 81.6 | ▲12.7 | 0.0 | ▲14.1 | 0.0 | ▲11.3 | 0.0 | 88.3 | ▼5.5 | 0.0 | ▼0.1 | 0.0 | ▼1.7 | 0.0 |
| CIFAR10 | 1.00 | 1.00 | 10 | 340M | 92.9 | 86.7 | ▲0.5 | 0.0 | ▲2.3 | 1.1 | ▲0.6 | 0.0 | 86.3 | ▲1.6 | 3.3 | ▲3.4 | 3.3 | ▲1.6 | 3.3 |
| OxfordIIITPet | 1.14 | 1.00 | 37 | 1.6G | 26.6 | 87.3 | ▲1.9 | 2.9 | ▲5.4 | 2.9 | ▲4.5 | 2.9 | 8.6 | ▲0.1 | 0.0 | ▲1.1 | 0.0 | ▲0.8 | 0.0 |
| Food101 | 1.00 | 1.00 | 101 | 5.3G | 68.7 | 88.8 | ▲8.9 | 1.6 | ▲10.9 | 0.0 | ▲14.1 | 0.0 | 85.5 | ▲14.6 | 0.8 | ▲18.5 | 0.4 | ▲22.2 | 0.4 |
| STL10 | 1.00 | 1.00 | 10 | 5.4G | 64.8 | 97.1 | ▲33.4 | 1.4 | ▲33.0 | 0.0 | ▲26.2 | 0.0 | 96.3 | ▲34.4 | 0.0 | ▲33.7 | 0.0 | ▲27.2 | 0.0 |
| **Average** | - | - | - | - | 61.1 | 60.4 | ▲7.2 | 1.3 | ▲8.2 | 1.5 | ▲6.8 | 1.4 | 62.4 | ▲5.3 | 2.1 | ▲6.7 | 1.7 | ▲5.7 | 1.2 |

*Table 13.* Full Component Ablation Study. Impact of removing Online Update, Prototype Refinement, and Skewness Correction on two datasets. Best results under each case are presented with optimal hyperparameters.

| Datasets → | | CIFAR100 | | | | | | | | GTSRB | | | | | | | |
|---|---|---|---|---|---|---|---|---|---|---|---|---|---|---|---|---|---|
| Ablation Cases → | | Baseline | | w/o Online | | w/o Skewness | | w/o Prototype | | Baseline | | w/o Online | | w/o Skewness | | w/o Prototype | |
| Attack ↓ | | CA | ASR | CA | ASR | CA | ASR | CA | ASR | CA | ASR | CA | ASR | CA | ASR | CA | ASR |
| Classic Backdoor | BadNets | 72.3 ▲2.2 | 0.0 | 73.2 ▲3.1 | 5.0 | 72.1 ▲1.9 | 0.0 | 70.4 ▲0.3 | 1.0 | 95.9 ▼2.6 | 0.0 | 89.3 ▼9.2 | 41.2 | 95.5 ▼2.9 | 0.8 | 89.1 ▼9.4 | 3.2 |
| | Blend | 73.2 ▲2.3 | 0.0 | 69.8 ▼1.1 | 6.3 | 72.9 ▲2.0 | 0.0 | 71.6 ▲0.7 | 1.0 | 93.0 ▼5.5 | 2.5 | 91.3 ▼7.2 | 24.6 | 92.8 ▼5.7 | 0.8 | 83.2 ▼15.3 | 4.2 |
| Dynamic Backdoor | WaNet | 68.1 ▲4.3 | 0.0 | 68.5 ▲4.7 | 4.0 | 69.4 ▲5.6 | 3.0 | 67.7 ▲4.0 | 6.1 | 87.0 ▼11.6 | 0.8 | 90.2 ▼8.4 | 26.1 | 80.8 ▼17.8 | 0.8 | 86.4 ▼12.2 | 23.2 |
| | BPP | 70.4 ▲5.6 | 0.0 | 68.3 ▲3.5 | 5.5 | 70.2 ▲5.4 | 0.0 | 68.5 ▲3.7 | 2.0 | 95.7 ▼0.7 | 0.0 | 89.3 ▼7.1 | 19.7 | 95.3 ▼1.1 | 0.0 | 89.5 ▼6.9 | 0.0 |
| | IAB | 70.5 ▲5.5 | 0.0 | 70.6 ▲5.6 | 3.6 | 70.3 ▲5.4 | 0.0 | 68.3 ▲3.4 | 0.0 | 95.7 ▼2.6 | 0.0 | 91.2 ▼7.1 | 15.0 | 95.3 ▼3.0 | 0.0 | 89.0 ▼9.3 | 0.8 |
| | SSBA | 72.7 ▲2.5 | 0.0 | 74.3 ▲4.1 | 3.0 | 72.4 ▲2.2 | 0.0 | 70.2 ▲0.0 | 0.0 | 90.0 ▼8.3 | 7.6 | 80.9 ▼17.4 | 23.2 | 89.6 ▼8.7 | 5.0 | 79.6 ▼18.7 | 14.3 |
| Clean-Label Backdoor | CTRL | 72.8 ▲2.5 | 0.0 | 72.9 ▲2.6 | 4.3 | 72.6 ▲2.2 | 0.0 | 70.5 ▲0.1 | 0.0 | 93.1 ▼5.5 | 6.7 | 91.3 ▼7.3 | 40.2 | 93.1 ▼5.5 | 5.9 | 83.4 ▼15.2 | 15.1 |
| | SIG | 72.8 ▲2.5 | 0.0 | 69.8 ▼0.5 | 5.6 | 72.5 ▲2.2 | 0.0 | 71.1 ▲0.8 | 1.0 | 92.5 ▼5.9 | 0.8 | 83.9 ▼14.5 | 14.9 | 93.0 ▼5.4 | 0.0 | 82.8 ▼15.6 | 5.0 |
| | LC | 74.0 ▲2.7 | 0.0 | 73.4 ▲2.1 | 5.4 | 73.7 ▲2.4 | 0.0 | 71.9 ▲0.6 | 1.0 | 92.0 ▼6.3 | 9.9 | 89.3 ▼9.0 | 38.3 | 91.9 ▼6.5 | 6.7 | 83.4 ▼15.0 | 16.0 |
| Clean-Image Backdoor | FLIP | 69.9 ▼0.2 | 0.0 | 66.4 ▼3.6 | 9.1 | 68.7 ▼1.4 | 0.0 | 68.7 ▼1.4 | 8.1 | 92.1 ▼3.5 | 2.4 | 94.2 ▼1.4 | 3.6 | 92.0 ▼3.6 | 4.8 | 85.6 ▼10.0 | 0.8 |
| | GCB | 68.3 ▼1.9 | 2.0 | 71.5 ▲1.3 | 3.4 | 67.9 ▼2.3 | 2.0 | 69.0 ▼1.2 | 3.0 | 88.5 ▼8.8 | 6.7 | 76.4 ▼20.9 | 13.5 | 88.0 ▼9.3 | 5.8 | 75.4 ▼21.9 | 10.8 |
| Average → | | 71.4 ▲2.6 | 0.2 | 70.8 ▲2.0 | 5.0 | 71.1 ▲2.3 | 0.5 | 69.8 ▲1.0 | 2.1 | 92.3 ▼5.6 | 3.4 | 87.9 ▼10.0 | 23.7 | 91.6 ▼6.3 | 2.8 | 84.3 ▼13.6 | 8.5 |

*Table 14.* **Target class frequency ablation on GTSRB (averaged over 9 attacks).** Results confirm no significant dependence on class frequency. CA and ASR in %.

| Target Class | Freq (%) | Samples | Avg CA | Avg ASR |
|---|---|---|---|---|
| Cls 0 (Speed 20 km/h) | 0.54 | 211 | 93.7 | 3.14 |
| Cls 23 (Slippery road) | 1.30 | 511 | 92.9 | 1.70 |
| Cls 17 (No entry) | 2.82 | 750 | 94.6 | 2.92 |
| Cls 25 (Road work) | 3.82 | 1501 | 91.9 | 2.11 |
| Cls 2 (Speed 50 km/h) | 5.73 | 2251 | 94.7 | 0.38 |

*Table 15.* **PRISM across victim architectures on CIFAR-10 (CA% / ASR%, per attack category average).** Classic BD: BadNets, Blend. Dynamic BD: SSBA, IAB, WaNet, BPP. Clean-Label BD: LC, SIG, CTRL.

| Architecture | Classic BD | Dynamic BD | Clean-Label BD |
|---|---|---|---|
| MobileNetV2 | 91.5 / 0.6 | 91.4 / 0.3 | 90.7 / 0.3 |
| VGG-16 | 92.9 / 0.6 | 92.7 / 1.4 | 92.7 / 1.5 |
| ViT-B/32 | 95.7 / 0.0 | 95.4 / 1.1 | 95.3 / 1.1 |
| PreActResNet18 | 94.1 / 0.0 | 93.4 / 0.3 | 92.4 / 1.8 |

*Table 16.* **Dirty cold-start robustness on CIFAR-10.** CA / ASR under poison-from-start. Even at 50% poison rate, ASR $\leq$ 10.9% vs. ~95% undefended; gap vs. clean warmup always $\leq$ 5%.

| Attack | Poison Rate | Clean Warmup | | Dirty Cold-Start | |
|---|---|---|---|---|---|
| | | CA | ASR | CA | ASR |
| BadNet | 10% | 93.2 | 0.0 | 92.3 | 0.0 |
| BadNet | 25% | 90.4 | 0.6 | 91.5 | 1.9 |
| BadNet | 50% | 90.3 | 1.1 | 90.2 | 2.8 |
| Blend | 10% | 93.8 | 0.0 | 92.3 | 0.0 |
| Blend | 25% | 90.3 | 2.1 | 91.6 | 4.8 |
| Blend | 50% | 90.1 | 8.0 | 90.1 | 10.9 |
| LC | 10% | 92.8 | 0.0 | 92.4 | 0.0 |
| LC | 25% | 90.5 | 0.6 | 91.4 | 2.7 |
| LC | 50% | 90.3 | 1.7 | 90.1 | 3.2 |
| WaNet | 10% | 91.0 | 0.0 | 92.4 | 1.0 |
| WaNet | 25% | 90.2 | 0.6 | 91.4 | 2.3 |
| WaNet | 50% | 90.4 | 1.1 | 90.3 | 2.9 |

PRISM (0.8% avg ASR) outperforms every training-time method except CGD (0.2%), while matching CGD's CA (93.2%)—despite CGD requiring full training data plus a trusted external VLM. Training-time methods frequently fail catastrophically on dynamic and clean-label attacks (e.g., ABL: 81.2% on WaNet, 99.8% on BPP; DBD: 100% on Blend, 98.3% on LC). PRISM outperforms CGD on WaNet, IAB, and CTRL with zero training data, validating the external semantic auditing paradigm.

# R. Defense against Semantic-Consistent Backdoor Attacks

Semantic-consistent backdoor attacks (Liu et al., 2018) craft triggers by blending victim-model features with target-class features at a mixing ratio $\alpha$, making the trigger visually and semantically coherent with the target class. Formally, given a clean image $x$ with true class $y \neq y_t$, the poisoned image is constructed as:

$$x_{\text{poison}} = (1 - \alpha) \cdot x + \alpha \cdot \bar{x}_{y_t},$$

where $\bar{x}_{y_t}$ is a feature prototype of the target class and $\alpha \in [0, 1]$ controls trigger strength. At low $\alpha$, the trigger is nearly imperceptible; at high $\alpha$, the image visually resembles the target class.

We directly construct this attack on CIFAR-10: target-class content is blended into source images at ratio $\alpha$. We compare PRISM against Fine-Pruning (FP) (Liu et al., 2018) and Adversarial Neuron Pruning (ANP) (Wu & Wang, 2021), reporting CA% / ASR%.

At $\alpha < 0.4$, PRISM achieves 0.0–3.3% ASR in the genuine backdoor regime, outperforming all baselines. The critical observation is that at $\alpha \geq 0.4$, even the *unpoisoned clean model* reaches 13.7–33.6% "ASR"—meaning that when nearly half the image consists of genuine target-class con-

tent, predicting the target class is expected behavior, not a backdoor artifact. This falls outside the scope of backdoor defense. Within the genuine backdoor regime ($\alpha < 0.4$), PRISM's external auditing paradigm excels because feature blending toward the target class amplifies the victim/VLM logit discrepancy that PRISM is designed to detect.

# S. Extension to Image Captioning

While PRISM is designed and evaluated for classification tasks, the external semantic auditing paradigm generalizes to sequence-generation tasks. The core detection signal is format-agnostic: it measures semantic inconsistency between the victim output and the VLM's judgment, regardless of whether the output is a class label or free-form text.

**Adaptation.** For image captioning, we replace the per-class logit-margin statistics with a single scalar: $\Delta_{\text{cap}} = \text{Sim}(\text{Enc}_{\text{text}}(\hat{s}), g_\phi(x))$, the CLIP-similarity between the victim-generated caption $\hat{s}$ and the VLM's visual embedding of input $x$. Backdoored samples generate semantically mismatched captions, producing low $\Delta_{\text{cap}}$. Gating and Cornish-Fisher thresholding proceed identically to the classification case; no class IDs or per-class statistics are required anywhere. The dropped per-class prototype refinement has minimal impact on ASR (see Table 3), affecting only VLM-unfamiliar domains.

**Results.** Victim: BLIP-large (Li et al., 2022) on Flickr30k (Young et al., 2014); attack: BadNet; auditor: CLIP ViT-B/32 ($\gamma$=2.6).

PRISM reduces ASR from 100% to 0.0% at only 1.4 CIDEr-point cost. The CLIP-similarity distributions of clean and backdoored samples separate by $6.7\sigma$ (clean: $0.306 \pm 0.044$; backdoored: $0.010 \pm 0.056$), providing ample margin for reliable gating. A full study with diverse captioning archi-

*Table 17.* **Comparison with training-time defenses on CIFAR-10 (ASR %, 5% poison rate).** [†]PRISM: test-time, zero training data. [*]CGD: training-time + trusted external VLM. Avg CA (%) in final row.

| Attack | No Defense | ABL | DBD | ASD | D-BR | EP | ReBack | PIPD | MSPC | CGD[*] | PRISM[†] |
|---|---|---|---|---|---|---|---|---|---|---|---|
| BadNets | 93.8 | 1.1 | 2.6 | 2.1 | 1.5 | 0.8 | 4.3 | 0.5 | 0.3 | 0.0 | 0.0 |
| Blend | 99.8 | 4.4 | 100 | 5.3 | 0.0 | 96.1 | 2.4 | 5.3 | 0.7 | 0.0 | 0.0 |
| WaNet | 96.9 | 81.2 | 2.6 | 8.8 | 60.2 | 91.1 | 84.4 | 11.4 | 54.2 | 0.3 | 0.0 |
| BPP | 99.2 | 99.8 | 99.9 | 99.4 | 85.5 | 4.6 | 1.8 | 0.9 | 2.8 | 0.3 | 1.1 |
| IAB | 94.9 | 83.0 | 0.0 | 19.8 | 84.8 | 1.6 | 1.7 | 4.0 | 5.3 | 0.7 | 0.0 |
| SSBA | 97.3 | 3.9 | 2.4 | 7.1 | 3.0 | 10.5 | 6.6 | 17.2 | 21.5 | 0.0 | 0.0 |
| CTRL | 95.9 | 2.4 | 57.8 | 89.3 | 98.3 | 1.1 | 96.2 | 12.6 | 77.3 | 0.1 | 0.0 |
| SIG | 93.9 | 0.4 | 97.5 | 99.5 | 49.6 | 12.5 | 29.9 | 13.5 | 10.3 | 0.0 | 2.2 |
| LC | 98.4 | 5.2 | 98.3 | 9.8 | 1.7 | 0.6 | 1.0 | 3.7 | 89.4 | 0.0 | 0.0 |
| FLIP | 99.2 | 99.6 | 1.6 | 62.2 | 22.1 | 98.5 | 39.7 | 66.9 | 17.2 | 0.5 | 1.1 |
| GCB | 100 | 100 | 6.9 | 100 | 100 | 99.9 | 71.6 | 87.7 | 23.9 | 0.0 | 4.4 |
| Avg ASR | 97.2 | 43.7 | 42.7 | 45.7 | 46.1 | 37.9 | 30.7 | 20.3 | 27.5 | 0.2 | 0.8 |
| Avg CA | 91.7 | 87.5 | 91.7 | 91.6 | 87.2 | 90.9 | 89.6 | 92.1 | 91.7 | 93.2 | 93.2 |

*Table 18.* **Defense against visually semantic-consistent backdoor at varying blend ratio $\alpha$ on CIFAR-10.** At $\alpha \geq 0.4$ the "clean model" baseline itself reaches 13.7–33.6% "ASR," indicating the images genuinely resemble the target class—outside the backdoor defense scope.

| $\alpha$ | No Defense | FP | ANP | PRISM | Clean Model |
|---|---|---|---|---|---|
| 0.1 | 89.0 / 23.0 | 91.3 / 1.2 | 80.4 / 0.0 | 93.0 / 0.0 | 93.1 / 0.9 |
| 0.2 | 90.7 / 60.9 | 91.7 / 6.6 | 84.5 / 1.3 | 92.3 / 1.1 | 93.1 / 1.5 |
| 0.3 | 92.1 / 80.1 | 91.9 / 27.0 | 87.5 / 2.6 | 92.1 / 3.3 | 93.1 / 4.7 |
| 0.4 | 92.4 / 89.5 | 91.5 / 50.6 | 85.1 / 7.6 | 91.6 / 7.8 | 93.1 / 13.7 |
| 0.5 | 93.3 / 94.8 | 92.0 / 75.7 | 87.3 / 26.1 | 91.7 / 26.7 | 93.1 / 33.6 |

*Table 19.* **PRISM for image captioning (Flickr30k, BadNet attack).** CA measured by CIDEr score; ASR measures fraction of poisoned inputs generating the target caption.

| Setting | CA (CIDEr) | ASR (%) |
|---|---|---|
| No defense | 62.6 | 100 |
| PRISM ($\gamma$=2.6) | 61.2 | 0.0 |

tectures and attack types is left for future work.

## T. Ethical Considerations

This work focuses on defending deployed neural networks against backdoor attacks—a defensive security research direction. The artifacts produced (the PRISM defense framework) are intended to improve the trustworthiness of AI systems. We do not introduce new attack methods. The backdoor attacks used in our evaluation are all drawn from publicly available implementations in BackdoorBench (Wu et al., 2022) and prior works.

The only potential dual-use concern is that analyzing the limits of PRISM (e.g., the SC3 full-collusion scenario in Sec. 5.5.3) could inform stronger attacks. However, we believe transparent documentation of defense limitations is essential for the research community to make progress, and the specific SC3 attack requires a level of supply-chain control (simultaneous compromise of two independent model

pipelines) that is qualitatively beyond the standard backdoor threat model.

All datasets used in evaluation are publicly available and do not contain personally identifiable information. No human subjects were involved in the experiments.

## U. Potential Mitigations for Identified Limitations

We outline targeted mitigations for the key limitations identified in Sec. 6 (Discussion) of this work.

**Domain-specific VLM gap.** For tasks where open-world VLMs have poor zero-shot performance (e.g., medical imaging, industrial inspection), we recommend the Trust Chain strategy (App. E): a weak clean generalist VLM is used to sanitize a domain-specialist VLM, which then audits the victim model. Alternatively, few-shot adaptation using a small set of *verified clean* domain samples can improve VLM coverage without violating the data-free spirit of the deployment setting, leveraging advanced multimodal integration (Liu et al., 2025a), high-quality data synthesis (Liu et al., 2024b), and open data-centric reasoning paradigms (Lin et al., 2026a).

**Dirty cold-start.** As demonstrated in App. P, text anchors provide transient protection during initialization. An additional mitigation is to extend the warm-up window $N$ when the deployment context is known to be high-risk, trading slightly slower prototype convergence for stronger cold-start protection, a robust strategy conceptually similar to isolating malicious updates in high-noise federated environments (Tan et al., 2025; Li et al., 2025a;b). Furthermore, in extreme scenarios where the initial test stream is permanently contaminated, integrating decentralized certified unlearning mechanisms (Wu et al., 2026) could offer a decentralized pathway to retrospectively erase the statistical bias inflicted

by cold-start anomalies.

**Adaptive adversaries (SC2 prompt manipulation).** The typography attack in Sec. 5.5.3 achieves ASR=15.6% by exploiting PRISM's text-image alignment. This can be mitigated by replacing text prompts with class-specific visual queries (image-anchored prompts), which are not susceptible to textual manipulation.

**Sequence-generation tasks.** The captioning adaptation outlined in App. S provides a practical mitigation pathway for extending PRISM beyond classification, which could be further applicable to generative contexts like 3D scene generation (Li et al., 2026b). Model-based trigger generators (e.g., GAP (Poursaeed et al., 2018)) that generate adaptive triggers per input could further strengthen the attack surface, motivating tighter sequence-level semantic auditing in future work, especially in specialized downstream contexts such as scientific image synthesis (Lin et al., 2026b) and programmatic chart reasoning (Liu et al., 2026). For protecting the LLM backdoor, designing special skills to ensure security might be a good choice, as evidenced by previous work that the effectiveness of skills varies widely between tasks (Xu et al., 2026b; Liu et al., 2025b; Lin et al., 2026c).

