# OpenReview forum: "From Internal Diagnosis to External Auditing: A VLM-Driven Paradigm for Data-Free Online Backdoor Defense"
_ICML.cc/2026/Conference — ICML 2026 regular_

### Official Review · Reviewer_r3x4 · 2026-03-04

**Soundness:** 2
**Presentation:** 3
**Significance:** 2
**Originality:** 2
**Overall Recommendation:** 3
**Confidence:** 4

**Summary:**

This paper addresses the vulnerability of deep neural networks to backdoor attacks by designing a complete framework for data-free online backdoor defense, PRISM, based on visual language models and statistical monitoring for prototype refinement and detection. In terms of research background and problems, existing data-free backdoor defense methods are all limited to the internal diagnosis paradigm and can be divided into two major categories: model repair and input robustness. The former relies on the internal attribute statistical features of the victim model and is easily bypassed by advanced dynamic attacks; the latter is based on the robustness assumption of input perturbation and is completely ineffective against semantic-level clean image attacks. Moreover, both types of methods are deeply coupled with the damaged parameters of the victim model, presenting fundamental security flaws.

**Compliance With Llm Reviewing Policy:**

Affirmed.

**Final Justification:**

After reading the authors' responses and the comments from other reviewers, I have decided to maintain the current score.

**Key Questions For Authors:**

(1) If the initial samples in the cold start phase are directionally contaminated, it will directly cause the initial values of the statistics and prototypes to deviate from the normal range, leading to the failure of the entire subsequent defense process. Has the author theoretically analyzed the attack surface of this selective update mechanism? For progressive poisoning and cold start initial contamination attacks, does the paper have corresponding defense mechanism designs and rigorous experimental verifications to ensure a secure closed loop in the online update process?

(2) Can the "external semantic audit" paradigm and PRISM framework proposed in the paper be transferred to these mainstream visual tasks? For scenarios such as open set and incremental learning, how to solve core problems such as the contamination of statistics by unknown category samples, the dynamic expansion of category prototypes, and the re-calibration of adaptive thresholds? Are there any preliminary theoretical analyses or experimental results to prove the cross-task generalization ability of the method?

(3) When the VLM itself is implanted with in-distribution backdoors or is targeted by adversarial samples, does the PRISM framework have systematic fault-tolerant mechanisms and robustness guarantees? Can the lower bound of the defense system's robustness regarding the security of the VLM itself be theoretically given?

**Limitations:**

yes

**Strengths And Weaknesses:**

**Strengths**

(1) Model repair methods are deeply coupled with the damaged parameters of the victim model, and dynamic attacks can bypass defenses by optimizing triggers to eliminate internal statistical traces. In this paper, a pre-trained general VLM is used as an independent and trusted third-party auditor, decoupling the defense mechanism from the weights of the victim model, theoretically eliminating the core attack surface of weight manipulation attacks.

(2) This paper does not simply use VLM for classification consistency verification, but designs solutions to address the two core pain points of VLM-based defense.

(3) Under a unified no-data testing setting, multiple SOTA defense baselines (4 model repair methods and 5 input robustness methods) were compared, and 11 mainstream attacks were evaluated, covering four categories: classic, dynamic, clean label, and clean image attacks. Special emphasis was placed on verifying clean image attacks and adaptive attacks, where traditional defenses completely fail.

**Weaknesses**

(1) The entire defense system of this paper is built on a core assumption: the high-confidence predictions of the victim model induced by backdoor triggers lack semantic support in the feature space of general VLMs. This assumption is only valid for patch-type, perturbation-type, and non-semantic triggers, but has a fundamental blind spot for the current state-of-the-art semantic-consistent backdoor attacks, which is the most critical limitation of the method.

(2) The paper only validates against text-level semantic misguidance attacks and has not conducted any evaluation, analysis, or defense design for visually semantic-consistent backdoor attacks, resulting in the defense capability of the method only covering traditional non-semantic backdoors and being unable to deal with semantic backdoors, a core development trend, leading to a serious lag in attack and defense.

(3) The online update mechanism of the paper is the core for achieving domain adaptation, but the security of this mechanism highly depends on two unstrictly verified premises: the initial samples in the cold start phase are clean and the samples passing through the gate are definitely clean. However, both of these premises do not hold in strong adversarial scenarios, leaving an exploitable attack surface for attackers.

(4) The entire methodological system of the paper is designed around the closed-set image classification task, from the definition of core metrics, module design to loss calculation, all highly rely on the predefined fixed category set and the logit output of the classification task, resulting in hard limitations on the application scenarios of the method and making it unable to be transferred to other mainstream visual tasks, with severe insufficiency in generalization.

(5) The entire defense system of the paper is built on the core assumption that "pre-trained general VLMs are absolutely clean, trustworthy, and cannot be manipulated as root trust nodes", but this assumption is fragile and insufficiently justified in real deployment scenarios, leaving the entire system at risk of single-point failure.

---

> ### Author Rebuttal · Authors · 2026-03-28
>
> We are encouraged that reviewers recognize the novelty of our method (FyFH,gk8x) and evaluation breadth (SEsB,gk8x,r3x4).
>
> ---
>
> ### **[W1+W2+Q3] Semantic backdoor & semantic-consistent backdoor.**
>
> **On W1.** Our paper **already evaluates two published semantic-trigger attacks**: GCB (visual attribute triggers: style, background, texture) — **4.4% ASR** while every baseline defense fails (Table 2); FMB (high-level semantic feature mixing trigger) — **≤2.2% ASR** (Table 8).
>
> **On W2.** We directly construct the attack the reviewer describes: a "visually semantic **consistent** backdoor" where **target-class content** blended into source images at ratio α (see [visualization](https://anonymous.4open.science/r/PRISM-552/rebuttal_experiments/semantic_blend_vis.png)). Results (CA/ASR%):
>
> |α|No Def|FP|ANP|PRISM|Clean Model|
> |-|-|-|-|-|-|
> |0.1|89.0/23.0|91.3/1.2|80.4/0.0|93.0/0.0|93.1/0.9|
> |0.2|90.7/60.9|91.7/6.6|84.5/1.3|92.3/1.1|93.1/1.5|
> |0.3|92.1/80.1|91.9/27.0|87.5/2.6|92.1/3.3|93.1/4.7|
> |0.4|92.4/89.5|91.5/50.6|85.1/7.6|91.6/7.8|93.1/13.7|
> |0.5|93.3/94.8|92.0/75.7|87.3/26.1|91.7/26.7|93.1/33.6|
>
> At α<0.4, PRISM achieves 0.0–3.3% ASR in the genuine backdoor regime, outperforming all baselines. At α≥0.4, even the **unpoisoned clean model** reaches 13.7–33.6% "ASR" — when nearly half the image is genuine class-B content, predicting class B is simply expected behavior, not a backdoor; this falls outside backdoor defense scope.
>
> ---
>
> ### **[W3+Q1] Cold-start contamination & gate convergence.**
>
> **Cold-start contamination.** Poisons in cold start cannot compromise our system. Before any prototype exists, text anchors take over: they don't support the backdoor-target prediction, so triggered samples fail the gate and cannot corrupt prototypes or statistics. **PRISM stays text-anchor-only until clean samples arrive**; warm-up (Appendix A.1) is only for performance, not security. Progressive poisoning is handled identically: per-batch filtering prevents poison accumulation.
>
> We add poison throughout stream from cold-start (Poison Rate = target-class fraction). Result shows that ASR ≤10.9% at 50%, gap vs. clean warm-up ≤5%:
>
> |Attack|Poison Rate|Clean Warmup|Dirty Cold-Start|
> |-|-|-|-|
> |BadNet|25%|90.4/0.6|91.5/1.9|
> |BadNet|50%|90.3/1.1|90.2/2.8|
> |Blend|25%|90.3/2.1|91.6/4.8|
> |Blend|50%|90.1/8.0|90.1/10.9|
> |LC|25%|90.5/0.6|91.4/2.7|
> |LC|50%|90.3/1.7|90.1/3.2|
> |WaNet|25%|90.2/0.6|91.4/2.3|
> |WaNet|50%|90.4/1.1|90.3/2.9|
>
> **Gate convergence.** The selective update mechanism **self-corrects** in both directions of τ: τ is anchored at the lower quantile of the admitted sample distribution (Cornish-Fisher expansion, Appendix A) and cannot drift arbitrarily.
>
> - **If τ is too permissive**: overlap-zone backdoor samples create O(ε) contamination; since triggered inputs have low Δ by design, the admitted set remains predominantly benign, and the perturbation is bounded and progressively diluted by CMA at 1/N rate (Appendix B), keeping statistics attracted to the clean distribution.
> - **If τ is too conservative**: prototypes self-heal as admitted clean samples accumulate, raising Δ for corner cases until they pass the gate.
>
> ---
>
> ### **[W4+Q2] Generalizability to other visual tasks.**
>
> The core detection signal is format-agnostic: it measures semantic inconsistency between victim output and VLM judgment, regardless of output type (class label or free-form text).
>
> We test on **image captioning** (open-ended generation). Victim: BLIP-large on Flickr30k; attack: BadNet; auditor: CLIP ViT-B/32. The only change from the classification version: replace per-class margin statistics with a single CLIP-similarity(image, caption) scalar, with no class IDs or per-class statistics anywhere.
>
> |Setting|CA(CIDEr)|ASR(%)|
> |-|-|-|
> |No defense|62.6|100|
> |PRISM (γ=2.6)|61.2|0.0|
>
> ASR 100%→0% at 1.4% CA cost; distributions separate by **6.7σ** (0.306±0.044 vs. 0.010±0.056). Key components require no classification-specific structure; the dropped per-class prototype refinement has minimal ASR impact (Table 3), affecting only VLM-unfamiliar domains (e.g., GTSRB). We will include this in the appendix.
>
> ---
>
> ### **[W5+Q3] VLM clean assumption.**
>
> Our assumption is considerably weaker than "VLM is absolutely clean." We only require VLM is not poisoned with the **same trigger toward the same target** as the victim, validated empirically:
>
> |Scenario|CA(%)|ASR(%)|
> |-|-|-|
> |Baseline (clean CLIP)|93.9|0.2|
> |Sc1: different trigger|93.6|0.8|
> |Sc2: same trigger, diff target|93.9|0.0|
> |Sc3: full collusion|93.7|100|
>
> Moreover, the only failure Sc3 is already addressed in Appendix E: **ML Trust Chain**, when the suitable VLM itself is untrusted. Defenders can use a very weak but trustworthy VLM to sanitize poisoned strong VLM, then use sanitized VLM to audit the victim, which recovers **95.8% CA and 4.5% ASR** under full collusion (Table 8).
>
> ---
>
> All experiments in rebuttal are open-sourced at: https://anonymous.4open.science/r/PRISM-552/rebuttal_experiments

---

> > ### Author Rebuttal · Reviewer_r3x4 · 2026-04-02
> >
> > After reading the authors' responses and the comments from other reviewers, I have decided to maintain the current score.

---

### Official Review · Reviewer_gk8x · 2026-03-04

**Soundness:** 3
**Presentation:** 2
**Significance:** 2
**Originality:** 2
**Overall Recommendation:** 4
**Confidence:** 4

**Summary:**

This paper proposes a novel data-free backdoor defense method called PRISM. PRISM is deployed at inference time and leverages an external vision–language model (VLM) as an auditing mechanism. While the victim model is used to classify benign inputs, existing VLMs (e.g., CLIP or Qwen-VL) assist in identifying poisoned samples and are adapted to the image classification task through the use of hybrid anchors. To enable implicit detection of poisoned examples, the authors introduce a statistically grounded approach based on the VLM’s output logits.

**Compliance With Llm Reviewing Policy:**

Affirmed.

**Final Justification:**

I find the paper borderline (due to the possibility of breaking the defense through the pretrained VLM), but I lean slightly towards acceptance due to the strengths written in the initial review.
The author's rebuttal changed my initial evaluation. I find it quite important that the rebuttal discussion on the backdooring of pretrained VLM is included in the main text of the next revision.

**Key Questions For Authors:**

1. L216, what would be an examples of the  predefined text anchors T?

**Limitations:**

The author have not discussed the limitations and the paper would benefit from including those (see Weaknesses).

**Strengths And Weaknesses:**

Strengths:
1. A novel approach to adapt vision-language models for domain specific task
2. Proposal of an adaptive threshold
3. Strong experimental section and results, testing against a variety of attacks using multiple models on many datasets

Weaknesses
1. There is a fundamental security vulnerability in using off-the-shelf vision-language model. As shown in the literature [1,2] both contrastively pretrained VLMs such as CLIP and open-ended generation models such as Qwen-VL or Llava can be poisoned or backdoored during pretraining or the instruction tuning process. If the VLM backdoor is activated using the same trigger as the victim classification model, the proposed approach may fail.
2. The use of a strong external VLM may significantly affect the clean accuracy. For example, linear probe on CLIP and DINO with ViT-G as backbone achieves 97-99% accuracy on CIFAR-10 [3]. Therefore, the comparison with existing baselines in the terms of clean accuracy is not valid.
3. Figure 1 and pages 6 and 7 are too dense. There are too many details inside Fig 1.

Comments:
1. L202-205 would benefit for a more formal explanation of a 'language modeling loss', 'candidate label token', ...
2. Figure 1 would benefit from a better caption
3. Figure 2 is missing a PRISM (qwen) on L302

[1] Carlini, N., & Terzis, A. (2021). Poisoning and backdooring contrastive learning. arXiv preprint arXiv:2106.09667.
[2] Liang, J., Liang, S., Liu, A., & Cao, X. (2025). Vl-trojan: Multimodal instruction backdoor attacks against autoregressive visual language models. International Journal of Computer Vision, 133(7), 3994-4013.
[3] Sabolić, I., Grcić, M., & Šegvić, S. (2025). Seal your backdoor with variational defense. In Proceedings of the IEEE/CVF International Conference on Computer Vision (pp. 752-764).

---

> ### Author Rebuttal · Authors · 2026-03-28
>
> We are encouraged that reviewers recognize the novelty of our method (FyFH, gk8x) and evaluation breadth (SEsB, gk8x, r3x4). We also thank the reviewer for bringing VIBE (Sabolić et al. [3]) to our attention; it is closely related work on using pre-trained models for defense, and we will discuss it in the final version.
>
> ---
>
> ### **[W1] VLM itself can be backdoored.**
>
> We agree this is a valid concern in principle. However, the reviewer’s scenario requires a stronger condition than stated: it is not sufficient for the VLM to share merely the **same trigger** — the VLM backdoor must also redirect toward the **same target** as the victim model, according to our experiment on BadCLIP.
>
> |Scenario|VLM Setup|CA(%)|ASR(%)|
> |-|-|-|-|
> |Baseline (clean CLIP)|no backdoor|93.9|0.2|
> |Sc1: different trigger|VLM compromises on a different trigger|93.6|0.8|
> |Sc2: same trigger, different target|VLM compromises → predicts class 1, victim predicts class 0|93.9|0.0|
> |Sc3: full collusion (same trigger + same target)|VLM compromises → also predicts class 0|93.7|100|
>
> This makes the threat surface considerably narrower: the attacker should (1) exploit a backdoor in an independently distributed, widely-audited foundation model (CLIP, Qwen-VL, etc.), and (2) use exactly the same trigger and target to poison the victim model. Such cross-model collusion is a significantly stronger adversarial assumption than existing mainstream backdoor attack literatures.
>
> Even so, **this worst case is already addressed in Appendix E**. Defender can use **ML Trust Chain** when the suitable VLM itself is untrusted. We construct a **Collusive Backdoor**: both the victim (ResNet) and a domain VLM (QuiltNet, medical imaging) are injected with the **exact same trigger toward same target**. The only trusted root is standard CLIP — clean, but weak (53.7% CA on 5-class-dataset LC25000). As expected, direct auditing with the compromised QuiltNet fails entirely (98.4% ASR). We uses CLIP to sanitize QuiltNet, then uses sanitized QuiltNet to audit the victim, which recovers **95.8% CA and 4.5% ASR** under full collusion:
>
> |Configuration|CA(%)|ASR(%)|
> |-|-|-|
> |Poisoned Victim (no defense)|98.7|98.3|
> |Trusted Generalist CLIP (clean, standalone)|53.7|0.6|
> |Poisoned Specialist QuiltNet (standalone)|75.2|98.1|
> |Direct Audit: Poisoned QuiltNet → Victim|96.4|98.4 ×|
> |Direct Audit: Clean CLIP → Victim|87.9|3.5|
> |**Trust Chain: CLIP → QuiltNet → Victim**|**95.8**|**4.5 ✓**|
>
> The defender needs only one trustworthy VLM — even a weak generalist suffices, because its pre-training is statistically independent of the specific attack trigger.
>
> ---
>
> ### **[W2] CA comparison fairness: VLM biases clean accuracy.**
>
> PRISM did **not** use a strong external VLM. We deliberately chose **CLIP ViT-B/32** with weight from OpenAI (the weakest VLM we know). It achieves only **86.4% zero-shot accuracy on CIFAR-10** and **26.6% on GTSRB** — far below the 97–99% accuracy in [3] from the reviewer, which uses large VLMs like DINO or ViT-G.
>
> Moreover, under PRISM's zero-annotation test-time threat model, linear probing is even not a feasible baseline: it requires labeled data. [3] also requires ground-truth training data to generate pseudolabels. Neither is applicable here.
>
> Most importantly, the baseline comparison in GTSRB (Table 7) directly refutes the concern: our VLM achieves only **26.6% zero-shot accuracy** on GTSRB, yet PRISM reaches **93.4% CA** — comparable to the average CA of existing baselines under the same threat model (93.8%). This shows that PRISM's high CA majorly comes from the system design, not from VLM classification capacity.
>
> Our CLIP ViT-B/32 capability:
> |Method|CIFAR-10 CA|GTSRB CA|Annotation Required|
> |-|-|-|-|
> |Zero-Shot CLIP|86.4%|26.6%|**None**|
> |Linear Probe|91.8%|69.3%|Test-set labels|
> |**PRISM (ours)**|**93.2%**|**93.4%**|**None**|
>
> ---
>
> ### **[W3] Presentation and Limitations.**
> We will address all presentation issues: (1) simplify Figure 1 by splitting into a high-level overview and a detailed component diagram; (2) add the missing PRISM(Qwen) annotation at L302 in Figure 2; (3) formalize "language modeling loss" and "candidate label token" at L202–205 with explicit mathematical definitions; (4) add a Limitations section covering the trust-chain requirement for specialized domains and the closed-set classification constraint; (5) standardize all figure and table caption formatting.
>
> ---
>
> **[Q1] Text anchor examples.** The predefined text anchor set T uses template + class_name prompts following the official CLIP prompt protocol. For CIFAR-10: T = {"a photo of an airplane", "a photo of an automobile", "a photo of a bird", ...}; for GTSRB: T = {"a photo of a speed limit 20 km/h sign", "a photo of a no entry sign", ...}, using the full template library from the CLIP official repository.
>
> ---
>
> To ensure reproducibility, all experiments added in this rebuttal are anonymously open-sourced at: https://anonymous.4open.science/r/PRISM-552/rebuttal_experiments

---

> > ### Author Rebuttal · Reviewer_gk8x · 2026-04-03
> >
> > I thank the authors for the strong rebuttal. I found my concerns addressed. I find it very important that the authors address the presentation issues as promised, as well as include the whole discussion on W1 in the next revision of the paper. At this point, I find the paper borderline (due to the possibility of breaking the defense through the pretrained VLM), but I lean slightly towards acceptance due to the strengths written in the initial review.

---

### Official Review · Reviewer_SEsB · 2026-03-09

**Soundness:** 3
**Presentation:** 3
**Significance:** 3
**Originality:** 3
**Overall Recommendation:** 5
**Confidence:** 3

**Summary:**

This paper proposes PRISM (Prototype Refinement & Inspection via Statistical Monitoring), a data-free test-time backdoor defense framework that shifts from internal diagnosis (model repairing, input robustness) to external semantic auditing using Vision-Language Models (VLMs). PRISM wraps a suspicious victim model with a dual-stream architecture: one stream processes inputs through the victim model, the other through a frozen VLM acting as an independent auditor. Three core components enable the defense: (1) a Hybrid VLM Teacher that fuses static text anchors with online-refined visual prototypes to bridge the domain gap; (2) an Adaptive Router using Cornish-Fisher expansion for skewness-aware thresholding on logit margins; (3) an online CMA-based update mechanism for prototype refinement and statistical monitoring.

**Compliance With Llm Reviewing Policy:**

Affirmed.

**Final Justification:**

Fully resolved - My concerns have been adequately addressed.

**Key Questions For Authors:**

Please kindly refer to the Weaknesses.

**Limitations:**

yes

**Strengths And Weaknesses:**

**Strengths:**

- **Comprehensive evaluation scope.** 17 datasets, 11 attack types (including challenging clean-image attacks FLIP/GCB), 8 defense baselines, and 6 VLM backbones. The breadth is exceptional for a defense paper.
- **Practical design choices.** O(1) memory update via CMA (no replay buffer), KV-cache optimization for generative VLMs, and competitive inference latency make deployment realistic.
- **Thorough ablation and robustness analysis.** Component ablation (Table 3) cleanly isolates contributions. Adaptive attack evaluation goes beyond standard threat models.

**Weaknesses:**

- **Cold-start vulnerability underexplored.** The warm-up phase uses only one batch (256 samples) with a Gaussian prior. No analysis quantifies defense quality during this critical initialization window: an adversary could concentrate poison samples in the first batch to corrupt prototype initialization before the CMA inertia takes effect.
- **Selective update creates a circular dependency.** Only samples passing the gate (Δ>τ) update statistics, but the gate itself depends on those statistics. If early thresholds are miscalibrated, the system could lock into a suboptimal state. No theoretical or empirical analysis of convergence guarantees is provided.
- **Missing comparison with training-time defenses.** While the threat model is data-free, comparing against training-time methods (even unfairly) would contextualize where test-time defense stands in the broader landscape.

---

> ### Author Rebuttal · Authors · 2026-03-28
>
> We thank the reviewer for the thorough and constructive review, and in particular for highlighting the practical deployment realism of PRISM. We address all raised concerns below.
>
> ---
>
> ### **[W1] Cold-start vulnerability.**
>
> Prototype corruption is self-prevented: prototypes only update when Δ > τ. Before any prototype exists, text anchors take over, and text anchors are likely to reject triggered samples with target label predictions. Thus triggered samples fail the gate and cannot corrupt prototypes. If the entire first batch is poisoned, prototypes simply remain as text anchors until clean samples arrive. The warm-up phase (Appendix A.1) exists only to accelerate this transition for performance, not as a security prerequisite.
>
> To directly stress-test this, we ran a new experiment where the test stream is poisoned from the very beginning (including cold-start). Here, poison rate = fraction of test-stream samples that are poisoned in the target class. Results (CA% / ASR%):
>
> |Attack|Poison Rate|Clean Warmup|Dirty Cold-Start|
> |-|-|-|-|
> |BadNet|10%|93.2/0.0|92.3/0.0|
> |BadNet|25%|90.4/0.6|91.5/1.9|
> |BadNet|50%|90.3/1.1|90.2/2.8|
> |Blend|10%|93.8/0.0|92.3/0.0|
> |Blend|25%|90.3/2.1|91.6/4.8|
> |Blend|50%|90.1/8.0|90.1/10.9|
> |LC|10%|92.8/0.0|92.4/0.0|
> |LC|25%|90.5/0.6|91.4/2.7|
> |LC|50%|90.3/1.7|90.1/3.2|
> |WaNet|10%|91.0/0.0|92.4/1.0|
> |WaNet|25%|90.2/0.6|91.4/2.3|
> |WaNet|50%|90.4/1.1|90.3/2.9|
>
> Even when all poison is front-loaded in the first batch at 50% poison rate, ASR stays ≤ 10.9% (vs. ~95% with no defense). The gap vs. clean warm-up is ≤ 5% in all cases.
>
> ---
>
> ### **[W2] Selective update: circular dependency and convergence.**
>
> The feedback loop is self-correcting in both directions. τ_k is always anchored at the lower quantile of the admitted sample distribution (Cornish-Fisher expansion, Appendix A), and is not a free parameter that can drift arbitrarily.
>
> - **If τ is too permissive**: overlap-zone backdoor samples create O(ε) contamination in the statistics. Since backdoor samples have low Δ and fail the gate by design, the admitted set is predominantly benign. This perturbation is therefore bounded and progressively diluted by CMA at rate 1/N (Appendix B), keeping the statistics attracted to the clean distribution.
> - **If τ is too conservative**: clean corner cases are temporarily over-rejected. As admitted clean samples accumulate, prototype refinement gradually improves the VLM’s representation of the domain, raising Δ for these corner cases until they pass the gate. The system self-heals through prototype refinement.
>
> ---
>
> ### **[W3] Comparison with training-time defenses.**
>
> We provide this comparison for contextualization, noting that the threat models differ: all listed baselines require full training data, while PRISM operates at test time with zero training data. Below is **ASR (%) on CIFAR-10 at 5% poison rate**:
>
> |Attack|NoDef|ABL|DBD|ASD|D-BR|EP|ReBack|PIPD|MSPC|CGD\*|**PRISM (Ours)†**|
> |-|-|-|-|-|-|-|-|-|-|-|-|
> |**Classic Backdoor**| | | | | | | | | | | |
> |BadNets|93.8|1.1|2.6|2.1|1.5|0.8|4.3|0.5|0.3|**0.0**|**0.0**|
> |Blend|99.8|4.4|100|5.3|**0.0**|96.1|2.4|5.3|0.7|**0.0**|**0.0**|
> |**Dynamic Backdoor**| | | | | | | | | | | |
> |WaNet|96.9|81.2|2.6|8.8|60.2|91.1|84.4|11.4|54.2|0.3|**0.0**|
> |BPP|99.2|99.8|99.9|99.4|85.5|4.6|1.8|0.9|2.8|**0.3**|1.1|
> |IAB|94.9|83.0|**0.0**|19.8|84.8|1.6|1.7|4.0|5.3|0.7|**0.0**|
> |SSBA|97.3|3.9|2.4|7.1|3.0|10.5|6.6|17.2|21.5|**0.0**|**0.0**|
> |**Clean-Label Backdoor**| | | | | | | | | | | |
> |CTRL|95.9|2.4|57.8|89.3|98.3|1.1|96.2|12.6|77.3|0.1|**0.0**|
> |SIG|93.9|0.4|97.5|99.5|49.6|12.5|29.9|13.5|10.3|**0.0**|2.2|
> |LC|98.4|5.2|98.3|9.8|1.7|0.6|1.0|3.7|89.4|**0.0**|**0.0**|
> |**Clean-Image Backdoor**| | | | | | | | | | | |
> |FLIP|99.2|99.6|1.6|62.2|22.1|98.5|39.7|66.9|17.2|**0.5**|1.1|
> |GCB|100|100|6.9|100|100|99.9|71.6|87.7|23.9|**0.0**|4.4|
> |**Avg ASR**|97.2|43.7|42.7|45.7|46.1|37.9|30.7|20.3|27.5|**0.2**|0.8|
> |**Avg CA**|91.7|87.5|91.7|91.6|87.2|90.9|89.6|92.1|91.7|**93.2**|**93.2**|
>
> \* CGD: training-time, requires full training data + a trusted external VLM. † PRISM (Ours): test-time, zero training data.
>
> Across all 11 attacks, PRISM (0.8% avg ASR) outperforms every training-time method except CGD (0.2%), while matching CGD's CA (93.2% vs 93.2%). CGD has full training data access, yet PRISM outperforms it on several attacks (WaNet, IAB, CTRL). Training-time methods frequently fail on dynamic and clean-label attacks (e.g., ABL: 81.2% on WaNet, 99.8% on BPP; DBD: 100% on Blend, 98.3% on LC). We will add this table to the appendix.
>
> ---
>
> To ensure reproducibility, all experiments added in this rebuttal are anonymously open-sourced at: https://anonymous.4open.science/r/PRISM-552/rebuttal_experiments

---

> > ### Author Rebuttal · Reviewer_SEsB · 2026-04-03
> >
> > Fully resolved - My concerns have been adequately addressed.

---

### Official Review · Reviewer_FyFH · 2026-03-12

**Soundness:** 4
**Presentation:** 3
**Significance:** 3
**Originality:** 4
**Overall Recommendation:** 5
**Confidence:** 4

**Summary:**

This paper proposes a new data-free backdoor defense (i.e., PRISM) based on external semantic auditing of universal vision-language models (VLMs). PRISM transforms external VLMs into independent auditors through online adaptation and statistical monitoring, thereby decoupling defenses from target victim models. The estimation results across 11 backdoor attacks, 17 datasets, and 6 backbone VLMs demonstrate the effectiveness and robustness of PRISM.

**Compliance With Llm Reviewing Policy:**

Affirmed.

**Final Justification:**

The rebuttal has fully addressed my main concerns.

**Key Questions For Authors:**

1. When evaluating the performance on imbalanced datasets, is the default target class a majority class or a minority class? Will the frequency of the target class in the dataset affect the defense performance? I hope the authors can provide the details and do some ablation studies on this.

2. Is it possible to evaluate the defense performance on other victim model architectures (e.g., VGG, DenseNet, ViT)?

**Limitations:**

yes

**Strengths And Weaknesses:**

**Strengths:**

- This paper proposes a novel defense strategy based on external semantic auditing of universal vision-language models (VLMs).

- The proposed PRISM method demonstrates strong empirical performance and robustness across diverse settings.

**Weaknesses:**

- The experimental setup details and evaluations on imbalanced datasets are insufficient. When evaluating the performance on imbalanced datasets, is the default target class a majority class or a minority class? Will the frequency of the target class in the dataset affect the defense performance? I hope the authors can provide the details and do some ablation studies on this.

- The evaluations have limited victim model diversity. All experiments use a single victim architecture (PreActResNet18). I would suggest the authors further evaluate the defense performance on other model architectures (e.g., VGG, DenseNet, ViT) to make the results more convincing.

Minor:

- The writing of this paper can be further improved. The manuscript contains several punctuation and capitalization inconsistencies. For example, ''clean'' on Page 1 should be corrected as ``clean''. In addition, Figure 3 and Table 3 use bold, capitalized captions, whereas other figures and tables do not follow the same style.

---

> ### Author Rebuttal · Authors · 2026-03-28
>
> We thank the reviewer for the positive assessment of our novelty and experimental robustness. We address all raised concerns below.
>
> ---
>
> ### **[W1] Imbalanced dataset: experimental details and ablation.**
>
> The default target class is **class 0**, a minority class at 0.54% of training data in GTSRB — class 0 is used uniformly across all datasets in the paper to avoid any appearance of cherry-picking. We conducted a systematic ablation over 5 target classes spanning a 10.7× frequency range, each evaluated under 9 attacks (FLIP and GCB excluded as they rely on pre-generated external files that cannot be re-run with an arbitrary target):
>
> |Target Class|Freq|Samples|Avg CA|Avg ASR|
> |-|-|-|-|-|
> |Cls 0 (Speed 20 km/h) ← **default**|0.54%|211|93.7%|3.14%|
> |Cls 23 (Slippery road)|1.30%|511|92.9%|1.70%|
> |Cls 17 (No entry)|2.82%|750|94.6%|2.92%|
> |Cls 25 (Road work)|3.82%|1501|91.9%|2.11%|
> |Cls 2 (Speed 50 km/h)|5.73%|2251|94.7%|0.38%|
>
> Results show that target class frequency has **no significant impact** on defense performance: ASR ranges from 0.38% to 3.14% with no statistically significant monotonic trend trend across a 10.7× frequency range. The highest ASR (Cls 0, 3.14%) and lowest ASR (Cls 2, 0.38%) both remain far below pre-defense baselines (67.8%–99.7%); variation across classes reflects specific target class semantics, not frequency. PRISM's detection signal is the logit margin between the VLM's support for the victim's prediction and the next best class. This signal is class-frequency-agnostic by design.
>
> ---
>
> ### **[W2] Victim architecture diversity.**
>
> Pre-defense ASRs ranged from 67.8% to 99.7% across all architectures and attack types. After applying PRISM (CA/ASR in %):
>
> |Model|Classic BD|Dynamic BD|Clean-Label BD|
> |-|-|-|-|
> |MobileNetV2|91.5 / 0.6|91.4 / 0.3|90.7 / 0.3|
> |VGG-16|92.9 / 0.6|92.7 / 1.4|92.7 / 1.5|
> |ViT-B/32|95.7 / 0.0|95.4 / 1.1|95.3 / 1.1|
> |PreActResNet18|94.1 / 0.0|93.4 / 0.3|92.4 / 1.8|
>
> PRISM reduces all per-category averaged ASRs to under 2% across every architecture and attack category. This architecture-agnosticism is a structural property of the dual-stream design: PRISM wraps the victim model externally and never accesses its internal parameters. The defense signal is computed entirely within the VLM branch, so its effectiveness depends on the VLM's feature space, not the victim's architecture.
>
> ---
>
> ### **[W3] Writing.**
>
> We will correct all punctuation and capitalization inconsistencies (including the `''clean''` issue on Page 1), standardize Figure 3 and Table 3 captions to match the rest of the document.
>
> ---
>
> To ensure reproducibility, all experiments added in this rebuttal are anonymously open-sourced at: https://anonymous.4open.science/r/PRISM-552/rebuttal_experiments

---

> > ### Author Rebuttal · Reviewer_FyFH · 2026-03-31
> >
> > The authors have thoroughly addressed my earlier concerns with clear and compelling experimental evidence. The paper now presents a technically solid contribution, supported by comprehensive defense evaluations across diverse settings. Accordingly, I am inclined to raise my score.

---

### Decision · Program_Chairs · 2026-04-30

**Decision:**

Accept (regular)

**Comment:**

This paper proposes PRISM, a data-free test-time backdoor defense that uses external VLMs as independent semantic auditors via online adaptation and statistical monitoring. Reviewer FyFH: The paper lacks sufficient experimental details on imbalanced datasets (e.g., target class frequency effects) and evaluates only a single victim architecture (PreActResNet18), limiting the demonstration of defense robustness. Reviewer SEsB: The defense exhibits underexplored cold-start vulnerability, a circular dependency in selective updates, and missing comparisons with training-time backdoor defenses. Reviewer gk8x: The approach assumes the VLM is trustworthy despite evidence that VLMs can be backdoored, uses a strong VLM that may bias clean accuracy comparisons, and presents overly dense figures and text. Reviewer r3x4: The method relies on an unverified core assumption that fails against semantic-consistent backdoor attacks, lacks evaluation for visual semantic backdoors, has an insecure online update mechanism, is limited to closed-set classification, and treats the VLM as a fragile root of trust.

Overall conclusion: The authors have thoroughly addressed all reviewer concerns through additional experiments, ablation studies, theoretical analysis, and clarifications; therefore, the paper is recommended for acceptance.